# Assessing the value of seasonal hydrological forecasts for improving water resource management: insights from a pilot application in the UK

Andres Peñuela[1], Christopher Hutton[2], Francesca Pianosi[1, 3]

[1]Civil Engineering, University of Bristol, Bristol, BS8 1TR, UK
[2]Wessex Water Services Ltd, Bath, BA2 7WW, UK
[3]Cabot Institute, University of Bristol, BS8 1UH, UK

*Correspondence to*: Andres Peñuela (andres.penuela-fernandez@bristol.ac.uk)

**Abstract.** Improved skill of long-range weather forecasts has motivated an increasing effort towards developing seasonal hydrological forecasting systems across Europe. Among other purposes, such forecasting systems are expected to support better water management decisions. In this paper we evaluate the potential use of a real-time optimisation system (RTOS) informed by seasonal forecasts in a water supply system in the UK. For this purpose, we simulate the performances of the RTOS fed by ECMWF seasonal forecasting systems (SEAS5) over the past ten years, and we compare them to a benchmark operation that mimics the common practices for reservoir operation in the UK. We also attempt to link the improvement of system performances, i.e. the forecast value, to the forecast skill (measured by the mean error and the Continuous Ranked Probability Skill Score) as well as to the bias correction of the meteorological forcing, the decision maker priorities, hydrological conditions and the forecast ensemble size. We find that in particular the decision maker priorities and the hydrological conditions exert a strong influence on the forecast skill-value relationship. For the (realistic) scenario where the decision-maker prioritises the water resource availability over energy cost reductions, we identify clear operational benefits from using seasonal forecasts, provided that forecast uncertainty is explicitly considered by optimising against an ensemble of 25 equiprobable forecasts. These operational benefits are also observed when the ensemble size is reduced up to a certain limit. However, when comparing the use of ECMWF-SEAS5 products to ensemble streamflow predictions (ESP), which are more easily derived from historical weather data, we find that ESP remains a hard-to-beat reference not only in terms of skill but also in terms of value.

## 1. Introduction

In a water-stressed world, where water demand and climate variability (Stocker et al., 2014) are increasing, it is essential to improve the efficiency of existing water infrastructure along with, or possibly in place of, developing new assets (Gleick, 2003). In the current information age there is a great opportunity to do this by improving the ways in which we use hydrological data and simulation models (the 'information infrastructure') to inform operational decisions (Gleick et al., 2013, Boucher et al., 2012).

Hydro-meteorological forecasting systems are a prominent example of information infrastructure that could be used to improve the efficiency of water infrastructure operation. The usefulness of hydrological forecasts has been demonstrated in several applications, particularly to enhance reservoir operations for flood management (Voisin et al., 2011, Wang et al., 2012, Ficchì et al., 2016) and hydropower production (Faber and Stedinger, 2001, Maurer and Lettenmaier, 2004, Alemu et al., 2010, Fan et al., 2016). In these types of systems, we usually find a strong relationship between the forecast skill (i.e. the forecast ability to anticipate future hydrological conditions) and the forecast value (i.e. the improvement in system performance obtained by using forecasts to inform operational decisions). However, this relationship becomes weaker for water supply systems, in which the storage buffering effect of surface and groundwater reservoirs may reduce the importance of the forecast skill (Anghileri et al., 2016, Turner et al., 2017) particularly when the reservoir capacity is large (Maurer and Lettenmaier, 2004, Turner et al., 2017). Moreover, in water supply systems, decisions are made by considering the hydrological conditions over lead time of several weeks or even months. Forecast products with such lead times, i.e. 'seasonal' forecasts, are typically less skilful compared to the short-range forecasts used for flood control or hydropower production applications.

When using seasonal hydrological scenarios or forecasts to assist water system operations, three main approaches are available: worst case scenario, ensemble streamflow prediction (ESP) and dynamical streamflow prediction (DSP). In the worst-case scenario approach, operational decisions are made by simulating their effects against a repeat of the worst hydrological droughts on records. Worst-case forecasts clearly have no particular skill, but their use has the advantage of providing a lower bound of system performance and reflect the risk-adverse attitude of most water management practice. This approach is commonly applied by water companies in the UK and it is reflected in the water resource management planning guidelines of the UK Environment Agency (EA, 2017).

In the ensemble streamflow prediction (ESP) approach, a hydrological forecasts ensemble is produced by forcing a hydrological model using the current initial hydrological conditions and historical weather data over the period of interest (Day, 1985). Operational decisions are then evaluated against the ensemble. The skill of the ESP ensemble is mainly due to the updating of the initial conditions. Since ESP forecasts are based on the range of past observations, they can have limited skill under non-stationary climate and where initial conditions do not dominate the seasonal hydrological response (Arnal et al., 2018). Nevertheless, the ESP approach is popular among operational agencies thanks to its simplicity, low cost, efficiency and its intuitively appealing nature (Bazile et al., 2017). Some previous studies assessed the potential of seasonal ESP to improve the operation of supply-hydropower systems. For example, Alemu et al. (2010) reported achieving an average economic benefit of 7% with respect to the benchmark operation policy, whereas Anghileri et al. (2016) reported no significant improvements (possibly because they only used the ESP mean, instead of the full ensemble).

Last, the dynamical streamflow prediction (DSP) approach uses numerical weather forecasts produced by a dynamic climate model to feed the hydrological model (instead of historical weather data). The output is also an ensemble of hydrological forecasts, whose skill comes from both the updated initial condition and the predictive ability of the numerical weather forecasts. The latter is due to global climate teleconnections such as the El Niño Southern Oscillation (ENSO) and the North Atlantic Oscillation (NAO). Therefore, DSP forecasts are generally more skilful in areas where climate teleconnections exert

a strong influence, such as tropical areas, and particularly in the first month ahead (Block and Rajagopalan, 2007). In areas where climate teleconnections have a weaker influence, DSP can have lower skill than ESP, particularly beyond the first lead month (Arnal et al., 2018, Greuell et al., 2019). Nevertheless, recent advances in the prediction of climate teleconnections in Europe, such as the NAO (Wang et al., 2017, Scaife et al., 2014, Svensson et al., 2015), means that seasonal forecasts skill is likely to continue increasing in the coming years. Post-processing techniques such as bias correction can also potentially

improve seasonal streamflow forecast skill (Crochemore et al., 2016). Studies assessing the benefits of bias correction for seasonal hydrological forecasting are still rare in the literature. While bias correction is often recommended or even required for impact assessments to improve forecast skills (Zalachori et al., 2012, Schepen et al., 2014, Ratri et al., 2019, Jabbari and Bae, 2020) studies on long-term hydrological projections (Ehret et al., 2012, Hagemann et al., 2011) highlighted a lack of clarity on whether bias correction should be applied or not. In recent years, meteorological centres such as the European Centre

for Medium-Range Weather Forecast (ECMWF) and the UK Met Office, have made important efforts to provide skilful seasonal forecasts, both meteorological (Hemri et al., 2014, MacLachlan et al., 2015) and hydrological (Bell et al., 2017, Arnal et al., 2018) in the UK and Europe, and encouraged their application for water resource management. To our knowledge, however, pilot applications demonstrating the value of such seasonal forecast products to improve operational decisions are mainly lacking and have only very recently started to appear (Giuliani et al., 2020).

While the skill of DSP is likely to keep increasing in the next years, it may still remain low at lead times relevant for the operation of water supply systems. Nevertheless, a number of studies have demonstrated that other factors, which are not necessarily captured by forecast skill scores, may also be important to improve the forecast value. These include accounting explicitly for the forecast uncertainty in the system operation optimization (Yao and Georgakakos, 2001, Boucher et al., 2012, Fan et al., 2016), using less rigid operation approaches (Yao and Georgakakos, 2001, Brown et al., 2015, Georgakakos and

Graham, 2008) and making optimal operational decisions during severe droughts (Turner et al., 2017, Giuliani et al., 2020). Additionally, the forecast skill itself can be defined in different ways, and it is likely that different characteristics of forecast errors (sign, amount, timing, etc.) affect the forecast value in different ways. Widely used skill scores for hydrological forecast ensembles are the rank histogram (Anderson, 1996), the relative operating characteristic (Mason, 1982) and the ranked probability score (Epstein, 1969). The ranked probability score is widely used by meteorological agencies since it provides a

measure of both the bias and the spread of the ensemble into a single factor, while it can also be decomposed into different sub-factors in order to look at the different attributes of the ensemble forecast (Pappenberger et al., 2015, Arnal et al., 2018). However, whether these skill score definitions are relevant for the specific purpose of water resources management, or whether other definitions would be better proxy of the forecast value, remains an open question.

In this paper, we aim at contributing to the ongoing discussion on the value of seasonal weather forecasts in decision making

(Bruno Soares et al., 2018) and at assessing the value of DSP for improving water system operation by application to a real-world reservoir system, and in doing so we build on the growing effort to improve seasonal hydrometeorological forecasting systems and make them suitable for operational use in the UK (Bell et al., 2017, Prudhomme et al., 2017). Through this application we aim to answer the three following questions: 1) can the efficiency of a UK real-world reservoir supply system

be improved by using DSP forecasts?, 2) does accounting explicitly for forecast uncertainty improve forecast value (for the same skill)? and 3) what other factors influence the forecast skill-value relationship?

For this purpose, we will simulate a real-time optimization system informed by seasonal weather forecasts over a historical period for which both observational and forecast datasets are available, and we will compare it to a worst-case scenario approach that mimics current system operation. As for the seasonal forecast products, we will assess both ESP and DSP derived from the ECMWF seasonal forecast products (Tim et al., 2018). We will also compare the forecast skill and value before and after applying bias correction, and for different ensemble sizes. System performances will be measured in terms of water availability and energy costs, and we will investigate five different scenarios for prioritising these two objectives depending on the decision-maker preferences. Finally, we will discuss opportunities and barriers to bring such approach into practice.

Our results are meant to provide water managers with an evaluation of the potential of using seasonal forecasts in the UK and to give forecasts providers indications on directions for future developments that may make their products more valuable for water management.

## 2. Methodology
### 2.1. Real-time optimization system

An overview of the real-time optimization system (RTOS) informed by seasonal weather forecasts is given in Figure 1 (left part). It consists of three main stages that are repeated each time an operational decision must be made. These three stages are:

1.a Forecast generation. We use a hydrological model forced by seasonal weather forecasts to generate the seasonal hydrological forecasts. The initial conditions are determined by forcing the same model by (recent) historical weather data for a warm-up period. Another model determines the future water demand during the forecast horizon. Although not tested in this study, in principle such a demand model could also be forced by seasonal weather forecasts.

1.b Optimization. This stage uses (i) a reservoir system model to simulate the reservoir storages in response to given inflows and operational decisions, (ii) a set of operation objective functions to evaluate the performance of the system, for instance, to maximize the resource availability or to minimize the operation costs, and (iii) a multi-objective optimizer to determine the optimal operational decisions. When a problem has multiple objectives, optimisation does not provide a single optimal solution (i.e. a single sequence of operational decisions over the forecast horizon) but rather it provides a set of (Pareto) optimal solutions, each realising a different trade-off between the conflicting objectives (for a definition of Pareto optimality see e.g. (Deb et al., 2002)).

1.c Selection of one trade-off solution. In this stage, we represent the performance of the optimal trade-off solutions in what we call a "pre-evaluation Pareto front". The terms "pre-evaluation" highlights that these are the anticipated performances according to our hydrometeorological forecasts, not the actual performances achieved when the decisions are implemented (which are unknown at this stage). By inspecting the pre-evaluation Pareto front, the operator will select one Pareto-optimal solution according to their priorities, i.e. the relative importance they give to each operation objective. In a simulation experiment, we can mimic the operator choice by setting some rule to choose one point on the Pareto front (and apply it consistently at each decision timestep of the simulation period).

## 2.2. Evaluation

When the RTOS is implemented in practice, the selected operational decision is applied to the real system and the RTOS used again, with updated system conditions, when a new decision needs to be made or new weather forecasts become available. If however we want to evaluate the performance of RTOS in a simulation experiment (for instance to demonstrate the value of using RTOS to reservoir operators) we need to combine it with the evaluation system depicted in the right part of Figure 1. Here, the selected operational decision coming out of the RTOS is applied to the reservoir system model, instead of the real system. The reservoir model is now forced by hydrological inputs observed in the (historical) simulation period, instead of the seasonal forecasts, which enables us to estimate the actual flows and next-step storage that would have occurred if the RTOS was used at the time. This simulated next-step storage can then be used as the initial storage volume for running the RTOS at the following timestep. Once the process has been repeated for the entire period of study, we can provide an overall evaluation of the hydrological forecast skill and the performance of the RTOS, i.e. the forecast value. This evaluation (Figure 1) consists of two stages:

2.a Forecast skill evaluation. The forecast skill is evaluated based on the differences between hydrological forecasts and observed reservoir inflows over the simulation period. For this purpose, we can calculate the absolute differences between the observed and the forecasted inflows or we can use forecast skill scores such as the continuous ranked probability skill score (CRPSS).

2.b Forecast value evaluation. The forecast value is presented as the improvement of the system performance obtained by using the RTOS over the simulation period, with respect to the performance under a simulated benchmark operation. Notice that, because the RTOS deals with multi objectives and hence provides a set of Pareto optimal solutions, in principle we could run a different simulation experiment for each point of the pre-evaluation Pareto front, i.e. for each possible definition of the operational priorities. However, for the sake of simplicity, we will simulate a smaller number of relevant and well differentiated operational priorities. The simulated performances of these solutions are visualised in a "post-evaluation" Pareto front. In this Pareto front diagram, the origin of the coordinates represents the performance of the benchmark operation, and the performances of any other solution are rescaled with respect to the benchmark performance. Therefore, a positive value along one axis represents an improvement in that operation objective with respect to the benchmark, whereas a negative value represents a deterioration. When values are positive on both axes, the simulated RTOS solution dominates (in a Pareto sense) the benchmark; the further away from the origin, the more the forecast has proven valuable for decision-making. If instead one value is positive and the other is negative then we would conclude that the forecast value is neither positive or negative, because the improvement of one objective was achieved at the expenses of the other.

## 2.3. Case study

### 2.3.1. Description of the reservoir system

The reservoir system used in this case study is a two-reservoir system in the South West of the UK (schematised in Figure 2). The two reservoirs are moderately sized with storage capacities in the order of 20,000,000 m$^3$ (S1) and 5,000,000 m$^3$ (S2) (the average of UK reservoirs is 1,377,000 m$^3$ (EA, 2017)). The gravity releases from reservoir S1 ($u_{S1,R}$) feed into river R, and

thus contribute to support downstream abstraction during low flows periods. Pumped releases from S1 ($u_{S1,D}$) and gravity releases from reservoir S2 ($u_{S2,D}$) are used to supply the demand node D. A key operational aspect of the system is the possibility of pumping water back from river R into reservoir S1. Pumped inflows ($u_{R,S1}$) may be operated in the winter months (from 1st November till 1st April) to supplement natural inflows, provided sufficient discharge is available in the river (R). This facility provides additional drought resilience by allowing the operator to increase reservoir storage in winter to help ensure that the demand in the following summer can be met. As the pump energy consumption is costly, there is an important trade-off between the operating cost of pump storage and drought resilience.

The pumped storage operation is constrained by a rule curve, and has operated in eleven years since 1995. The rule curve defines the storage level at which pumps are triggered. Each point on the curve is derived based on the amount of pumping that would be required to fill the reservoir by the end of the pump storage period (1st April), under the worst historical inflows scenario. The pumping trigger is therefore risk-averse, which means there is a reasonable chance of pumping too early on during the refill period and increasing the likelihood of reservoir spills if spring rainfall is abundant. This may result in unnecessary expenditure on pumping. Informing pump operation by using seasonal forecasts of future natural inflows ($I_{S1}$ and $I_{S2}$) may thus help to reduce the volume of water pumped whilst achieving the same reservoir storage at the end of the refilling period.

### 2.3.2. Forecast generation

In this study we generated dynamical streamflow predictions (DSP) by forcing a lumped hydrological model, the HBV model (Bergström and Singh, 1995), with the seasonal ECMWF SEAS5 weather hindcasts (Tim et al., 2018, Johnson et al., 2019). The ECMWF SEAS5 hindcast dataset consists of an ensemble of 25 members starting on the 1st day of every month and providing daily temperature and precipitation with a lead time of 7 months. The spatial resolution is 36 km which compared to the catchment sizes (28.8 $km^2$ for S1 and 18.2 $km^2$ for S2) makes it necessary to downscale the ECMWF hindcasts. Given the lack of clarity in the potential benefits of bias correction (Ehret et al., 2012), we will provide results of using both non-corrected and bias corrected forecasts. The dataset of weather hindcast is available from 1981, whereas reservoir data are available for the period 2005-2016. Hence, we used the period 2005-2016 for the RTOS evaluation and the earlier data from 1981 for bias correction of the meteorological forcing. While limited, this period captures a variety of hydrological conditions, including dry winters in 2005-06, 2010-2011 and 2011-12, which are close to the driest period on records (1975-1976) (see more details in Figure 7 of the Supplementary Material). This is important because, under drier conditions, the system performance is more likely to depend on the forecast skill and the benefits of RTOS may become more apparent (Turner et al., 2017). Daily inflows were converted to weekly inflows for consistency with the weekly time step applied in the reservoir system model.

A linear scaling approach (or "monthly mean correction") was applied for bias correction of precipitation and temperature forecasts. This approach is simple and often provides similar results in terms of bias removal as more sophisticated approaches such as the quantile or distribution mapping (Crochemore et al., 2016). A correction factor is calculated as the ratio (for precipitation) or the difference (for temperature) between the average daily observed value and the forecasted value (ensemble

mean), for a given month and year. The correction factor is then applied as a multiplicative factor (precipitation) or as an additive factor (temperature) to correct the raw daily forecasts. A different factor is calculated and applied for each month and each year of the evaluation period (2005-2016). For example, for November 2005 we obtain the precipitation correction factor as the ratio between the mean observed rainfall in November from 1981 to 2004 (i.e. the average of 24 values) and the mean forecasted rainfall for those months (i.e. the average of 24x25 values, as we have 25 ensemble members). For November 2006, we re-calculate the correction factor by also including the observations and forecasts of November 2005, hence taking averages over 25 values; and so forth. The rationale of this approach is to best mimic what would happen in real-time, when the operator would likely access all the available past data and hindcasts for the bias correction.

As anticipated in the Introduction, the ESP is an ensemble of equiprobable weekly streamflow forecasts generated by the hydrological model (HBV in our case) forced by meteorological inputs (precipitation and temperature) observed in the past. For consistency with the bias correction approach used for the ECMWF SEAS5 hindcasts, we produce the ESP using meteorological observations from 1981 until the year before the simulated decision timestep. This leads to producing an ensemble of increasing size (from 24 to 35 members) but roughly similar to the ECMWF ensemble size (25 members).

### 2.3.3. Optimization: Reservoir system model, objective functions and optimiser

The reservoir system dynamics is simulated by a mass balance model implemented in Python. The simulation model is linked to an optimiser to determine the optimal scheduling of pumping ($u_{R,S1}$) and release ($u_{S1,D}$ and $u_{S2,D}$) decisions. For the optimiser we use the NSGA-II multi-objective evolutionary algorithm (Deb et al., 2002) implemented in the open-source Python package Platypus (Hadka, 2018). We set two operation objectives for the optimiser: to minimize the overall pumping energy cost and to maximize the water resource availability at the end of the pump storage period. The pumping cost is calculated as the sum of the weekly energy costs associated to pumped inflows and pumped releases ($u_{R,S1}$ and $u_{S1,D}$) over the optimisation period. The resource availability is the mean storage volume in S1 and in S2 at the end of the optimisation period (1$^{st}$ April). When optimisation is run against a forecast ensemble, the two objective functions are evaluated against each ensemble member and the average is taken as final objective function value. The gravity releases from S1 ($u_{S1,R}$) are not considered as decision variables and they are set to the observed values during the period of study. This choice is unlikely to have important implications on the optimization results because $u_{S1,R}$ on average represents only 15% of the total releases from S1 ($u_{S1,D}$ + $u_{S1,R}$). Also, we assume that future water demands are perfectly known in advance, and set them to the sum of the observed releases from S1 ($u_{S1,D}$) and S2 ($u_{S2,D}$) for the period of study. This simplification is reasonable for our case study as the water demand is fairly stable and predictable in winter, and it enables us to focus on the relationship between skill and value of the seasonal hydrological forecasts while assuming no error in demand forecasts. More details about the reservoir simulation model and the optimisation problem are given in the Supplementary Material.

### 2.3.4. Selection of the trade-off solution

We use five different rules for the selection of the trade-off solution from the pre-evaluation Pareto front (see Figure 1), and apply them consistently at each decision timestep of the simulation period. The five rules correspond to five different scenarios of operational priorities. They are: 1) resource availability only (*rao*), which assumes that the operator consistently selects the

extreme solution that delivers the largest improvement in resource availability; 2) resource availability prioritised (*rap*) selects the solution delivering the 75% percentile in resource availability increase; 3) balanced (*bal*) selects the solution delivering the median improvement in resource availability; 4) pumping savings prioritised (*psp*) selects the solution delivering the 75% percentile in energy cost reductions; and 5) pumping savings only (*pso*), which selects the best solution for energy saving.

### 2.3.5. Forecast skill evaluation

We use two metrics, a skill score and the mean error, to evaluate the quality of the hydrological forecasts over our simulation period (from November to April).

A skill score evaluates the performance of a given forecasting system with respect to the performance of a reference forecasting system. As a measure of performance, we use the continuous ranked probability score (CRPS) (Brown, 1974) (Hersbach, 2000). The CRPS is defined as the distance between the cumulative distribution function of the probabilistic forecast and the

empirical distribution of the corresponding observation. At each forecasting step, the CRPS is thus calculated as:

$$CRPS(p(x), I^{Obs}) = \int (p(x) - H(x < I^{Obs}))^2 dx$$

where *p(x)* represents the distribution of the forecast; $I^{Obs}$ is the observed inflow [m³]; and $H$ is the empirical distribution of the observation, i.e. the step function which equals 0 when $x < I^{Obs}$ and 1 when $x > I^{Obs}$. The lower the CRPS, the better the performance of the forecast. In this study weekly forecast and observation data were used to compute individual CRPS values.

The skill score is then defined as:

$$CRPSS = 1 - \frac{CRPS^{Sys}}{CRPS^{Ref}}$$

When the skill score is higher (lower) than zero, the forecasting system is more (less) skilful than the reference. When it is equal to zero, the system and the reference have equivalent skill. Following the recommendation by Harrigan et al. (2018) we used ensemble streamflow predictions (ESP) as a "tough to beat" reference, which is more likely to demonstrate the "real skill"

of the hydrological forecasting system (Pappenberger et al., 2015) based on dynamic weather forecasts.

The mean error measures the difference between the forecasted and the observed inflows (at monthly scale). The mean error is negative when the forecasts tend to underestimate the observations and positive when the forecasts overestimate the observations. The mean error for a given forecasting step and lead time *T* [months] is:

$$mean\ error = \frac{1}{M} \sum_{m=0}^{M} \left( \frac{1}{T} \sum_{t=0}^{T} (I_{t,m}^{Sys} - I_t^{Obs}) \right)$$

where *I* is the inflow [m³], *t* is the timestep [month] and *M* the total number of members (*m*) of the ensemble.

### 2.3.6. Forecast value evaluation and definition of the benchmark operation

To evaluate the value of the hydrological forecasts, we compared the simulated performance of the RTOS informed by these forecasts with the simulated performance of a benchmark operation. The benchmark mimics common practices in reservoir operation in the UK, whereby operational decisions are made against a worst-case scenario – a repeat of the worst hydrological

drought on records (1975-76). This comparison enables us to show the potential benefits of using seasonal forecast with respect

to the current approach. We simulate the benchmark operation using similar steps as in the RTOS represented in Figure 1, but with three main variations. First, instead of seasonal weather forecasts, we use the historical weather data recorded in Nov 1975-Apr 1976 (the worst drought on records). Second, the optimiser determines the optimal scheduling of reservoir releases ($u_{S1,D}$ and $u_{S2,D}$) but not that of pumped inflows ($u_{R,S1}$). Instead, these are determined by the rule curve applied in the current operation procedures. Specifically, if at the start of the week the storage level in S1 is below the storage volume defined by the rule curve for that calendar day, the operation triggers the pumping system during that week (we assume that the triggered pumped inflow is equal to the maximum pipe capacity). Third, the optimiser only aims at minimising pumping costs, whereas the resource availability objective is turned into a constraint, i.e. the mean storage volume of the two reservoirs must be maximum by the end of the pump storage period (1st April) and no trading-off with pumping costs reduction is allowed.

## 3.  Results

### 3.1  Forecast skills

First, we analyse the skill of DSP hydrological forecasts. Figure 3a shows the average CRPSS at different lead times before (red) and after (blue) bias correction of the meteorological forecasts. We compute the average CRPSS for a given lead time as the average of the CRPSS obtained for each forecast used for the simulation of the reservoir system in that time frame. For instance, the forecast for a 3-month lead time, since the simulation time frame is in this case 1 Jan to 1 Apr, we average the CRPSS values obtained for the 1 Jan-1 Apr, 1 Feb-1 Apr and 1 Mar-1 Apr forecasts.

Before bias correction, the average skill score is positive, i.e. the forecast is more skilful than the benchmark (ESP), only at 1-month or 2-month lead times (solid red line). CRPSS is higher than average in the three driest winters, i.e. 2005-2006, 2010-2011, 2011-2012 (dashed lines). If we compare DSP to DSP-corr (red and blue solid lines), we see that bias correction deteriorates the average skill scores for shorter lead times (1 and 2 months) while it improves them for longer ones (3,4 and 5 months) but the value is still negative, i.e. the forecast is less skilful than the benchmark (ESP). In the driest years (dashed lines) bias correction deteriorates the skill score for most lead times.

We compute the average mean error for a given lead time as for CRPSS. The computed mean error values (Figure 3b) indicate that DSP systematically underestimates the inflow observations but less so in the three driest winters. After bias correction (DSP-corr), this systematic underestimation turns into a systematic overestimation. Also, the average mean error gets lower for longer lead times, though not as much in the driest years.

In summary, we can conclude that bias correction does not seem to produce an improvement in the forecast skill for our observation period. On the other hand, what we find in our case study is a clear signal of bias correction turning negative mean errors (inflow underestimation) into positive errors (overestimation). So, while the magnitude of errors stays relatively similar, the sign of those errors changes. We will go back to this point later on, when analysing the skill-value relationship.

### 3.2  Forecast value

The forecast value is presented here as the simulated system performance improvement, i.e. increase in resource availability and in pumping cost savings, with respect to the benchmark operation.

### 3.2.1 Effect of operational priority scenario and forecast product on the forecast value

We start by analysing the average forecast value over the simulation period 2005-2016 (Figure 4) for the three seasonal forecast products (DSP, DSP-corr and ESP) and the perfect forecast, under five operational policy scenarios (rao: resource availability only; rap: resource availability prioritised; bal: balanced; psp: pumping savings prioritised; and pso: pumping savings only). Firstly, we notice in Figure 4 that the monthly pumping energy cost savings vary widely with the operational priority. The range of variation depends on the forecast type, going from £20,000 to £48,000 for the perfect forecast and from -£77,000 to £48,000 for DSP, DSP-corr and ESP. For all forecast products, the improvement in resource availability shows lower variability, with an improvement of less than +2% (of the mean storage volume in S1 and in S2 at the end of the optimisation period) for *rao*, and a deterioration of -2% for *pso*. While this seems to suggest a lower sensitivity of the resource availability objective, variations of few percent points in storage volume may still be important in critically dry years.

As for the forecast value, we find that the perfect forecast brings value (i.e. a simultaneous improvement of both objectives) in the two scenarios that prioritize the increase in resource availability (*rao* and *rap*), DSP brings no value in any scenarios, DSP-corr has positive value in the *rap* and *bal* scenario, and ESP in the *bal* only. In other words, real-time optimisation based on seasonal forecasts can outperform the benchmark operation, but whether this happens depends on both the forecast product being used and the operational priority.

An interesting observation in Figure 4 is that the distance in performance between using perfect forecasts and real forecasts (DSP, DSP-corr, ESP) is very small under scenarios that prioritise energy savings (bottom-right quadrant) and much larger under scenarios prioritising resource availability (top quadrants). This indicates a stronger skill-value relationship under the latter scenarios, i.e. improvements in the forecast skill are more likely to produce improvements in the forecast value if resource availability is the priority.

Last, if we compare DSP with DSP-corr we see that the effect of bias correcting the meteorological forcing is mainly a systematic shift to the right along the horizontal axis, i.e. an improvement in energy cost savings at almost equivalent resource availability. Thanks to this shift, in the scenario that prioritises resource availability (*rap*), DSP-corr outperforms ESP. In fact, using DSP-corr is win-win with respect to the benchmark (i.e. the *rap* performance falls in the top-right quadrant in Figure 4) while using ESP is not, as it improves the resource availability at the expenses of pumping energy savings (i.e. producing negative savings).

### 3.2.2 Effect of the forecast ensemble size on the forecast value

We now analyse the effect that different characterisations of the forecast uncertainty have on the DPS-corr forecast value. We start by the extreme case when uncertainty is not considered at all in the real-time optimisation, i.e. when we take the mean value of the DSP-corr forecast ensemble and use it to drive a deterministic optimisation. The results are reported in Figure 5, which shows that the solution space shrinks to the bottom-right quadrant and, no matter the decision maker priority, the deterministic forecast has no value because energy savings are only achieved at the expenses of reducing the resource availability.

We also consider intermediate cases where optimisation explicitly considers the forecast uncertainty (i.e. it is based on the
average value of the objective functions across a forecast ensemble) but the size of the ensemble varies between 5 and 25
members (the original ensemble size). For clarity of illustration, we focus on the resource availability prioritised (*rap*) scenario
only. We choose this scenario because it seems to best reflect the current preferences of the system managers, whose priority
is to maintain the resource availability while reducing pumping costs as a secondary objective. Moreover, the previous analysis
(Figure 4) has shown that the optimised *rap* has a larger window of opportunity for improving performance with respect to the
benchmark and could potentially improve both operation objectives if the forecast skill was perfect.

For each chosen ensemble size, we randomly choose 10 replicates of that same size from the original ensemble, then we run a
simulation experiment using each of these replicates, and finally average their performance. Results are again shown in Figure
5. For a range of 10 to 20 ensemble members, the forecast value remains relatively close to the value obtained by considering
the whole ensemble (25 members). However, if only 5 members are considered, the resource availability is definitely lower
and cost savings higher, so that the trade-off that is actually achieved is different from the one that was pursued (i.e. to prioritise
resource availability). Notice that the extreme case of using 1 member, i.e. the deterministic forecast case (green cross in Figure
5), further exacerbates this effect of 'achieving the wrong trade-off' as resource availability is even lower than in the
benchmark.

### 3.2.3    Year-by-year analysis of the forecast value

Last, we investigate the temporal distribution of the forecast skill and value (i.e. increased resource availability and energy
cost savings) along the simulation period and compare it to the hydrological conditions observed in each year (Figure 6). The
"hydrological conditions" is the sum of the initial storage value and the total inflows during the optimisation period, hence
enabling us to distinguish dry and wet years. Again, for the sake of simplicity we focus on the simulation results in the most
relevant priority scenario of resource availability priority (rap). First, we observe that two specific years play the most
important role in improving the system performance with respect to the benchmark: 2010-11 for pumping cost savings (Figure
6e) and 2011-12 for resource availability (Figure 6d). These years correspond to the driest conditions in the period of study
(see Figure 6a , and the Supplementary Material for further analysis of the inflow data) but not to the highest forecast skills
either quantified with CRPSS or mean error (Figure 6b and c). In general, the temporal distribution of the average yearly
forecast skill does not show any correspondence with the yearly forecast value. When comparing DSP-corr with DSP (blue
and grey bars), we observe that they perform similarly in terms of resource availability but DSP-corr performs better for energy
savings. This difference was observed already when looking at average performances over the simulation period (Figure 4)
and can be related to the change in sign of forecasting errors induced by the bias correction of the meteorological forcing
(Figure 3b). In fact, without bias correction, reservoir inflows tend to be underestimated, which leads the RTOS to pump more
frequently and often unnecessarily (e.g. in 2005-06, 2006-07, 2007-08, etc.). With bias correction, instead, inflows tend to be
overestimated, and the RTOS uses pumping less frequently. Interestingly, the reduction in pumping still does not prevent to
improve the resource availability with respect to the benchmark. This is achieved by the RTOS through a better allocation of
pump and release volumes over the optimisation period. When comparing DSP-corr with ESP, we find that the largest

improvements are gained in the same years by both products, i.e. in the driest ones. As already emerged from the analysis of

average performances (Figure 4), we see that ESP achieves slightly better resource availability than DSP-corr but with less pumping cost savings. ESP in particular seems to produce 'unnecessary' pumping costs in 2006-07, 2011-12 and 2013-14, where DSP-corr achieves a similar resource availability (Figure 6d) at almost no cost (Figure 6e). It must be noted that for the ESP approach, these three specific years, 2006-07, 2011-12 and 2013-14. play the most important role in decreasing the pumping energy cost savings with respect to the benchmark.

## 4. Discussion

Our study provides some insights on the complex relationship between forecast skill and its value for decision-making. Although these findings may be dependent on the case study and time period that was available for the analysis, they still enable us to draw some more general lessons that could be useful also beyond the specific case investigated here.

First, we found that the use of bias correction, and in particular linear scaling of the meteorological forcing, to improve the

skill and value of DSP forecast is less straightforward than possibly expected. Our results show that on average bias correction does not improve the DSP forecast skill (as measured by CRPSS and mean error) and can even deteriorate it in dry years (Figure 3). This is because in our system DSP forecasts systematically underestimate inflows (before bias correction), which means their skill is relatively higher in exceptionally dry years and is deteriorated by bias correction. To our knowledge, no previous study reported such difference in skill for the ECMWF SEAS5 forecasts in dry years in the UK, hence we are not

able to say whether our result applies to other systems in the region. However, the result points at a possible intrinsic contradiction in the very idea of bias correcting based on climatology. In this study, the main reason for the bias correction to fail in improving the forecast skills is that the DSP forecast before bias correction was already performing relatively well in terms of skills in the three particularly dry winters (Figure 3) and worse in the rest, which are less dry and hence closer to the average climate conditions. After bias correction we worsened the forecast skills of these three exceptionally dry winters, but

we improved the skills in the rest. In this case the bias correction would have performed better if these three dry years were not considered, i.e. under less exceptional climate conditions the bias correction would have been more effective. More generally, by pushing forecasts to be more alike climatology, one may reduce the 'good signal' that may be present in the original forecast in years that will indeed be significantly drier (or wetter) than climatology. As exceptional conditions are likely the ones when water managers can extract more value from forecasts, the argument that bias correction ensures average

performance at least equivalent to climatology or ESP (e.g. Crochemore et al. (2016)) may not be very relevant here. We would conclude that more studies are needed to investigate the benefits of bias correction when seasonal hydrological forecasts are specifically used to inform water resource management.

While we could not find an obvious and significant improvement of forecast skill after bias correction, we found a clear increase in forecast value (Figure 4). RTOS based on bias-corrected DSP considerably reduces pumping costs with respect to

the original DSP, while ensuring similar resource availability. A consequence of this is that decision maker priorities *rap* (resource availability prioritised) and *bal* (balanced) dominate (in a Pareto sense) the benchmark. We explained this reduction

in pumping costs by the change in the sign of forecasting errors induced by bias correction – from a systematic underestimation of inflows to a systematic overestimation. While this change is again case specific, a general implication is that not all forecast errors have the same impact on the forecast value, and thus not all skill scores may be equally useful and relevant for water resource managers. For example, in our case a score that is able to differentiate between overestimation and underestimation errors, such as the mean error, seems more adequate than a score such as CRPSS, which is insensitive to the error sign. This said, our results overall suggest that inferring the forecast value from its skill may be misleading, given the weak relationship between the two (at least as long as we use skill scores that are not specifically tailored to water resources management). Running simulation experiments of the system operation, as done in this study, can shed more light on the value of different forecast products.

While we found a weak relationship between forecast skill and value, we found that forecast value is more strongly linked to hydrological conditions (Figure 6). As expected, a forecast-based RTOS system is particularly useful in dry years, where we find most of the gains with respect to the benchmark operation (Figure 6). This is consistent with previous studies for water supply system, e.g. Turner et al., 2017. In our case study, RTOS not only improve resource availability but also reduce pumping costs because, in the dryer years, storage levels are more likely to cross the rule curve and trigger pumping in the benchmark operation.

In light of the pre-processing costs of seasonal weather forecasts, it is interesting to discuss whether their use is justified with respect to a possibly simpler-to-use product such as ESP. While weather forecast centres are increasingly reducing the pre-processing costs by facilitating access to their seasonal weather forecast datasets, bias correction still needs a considerable level of expertise. This is not only because the necessary tools are currently not provided but also because we first need the knowledge to decide whether applying bias correction is appropriate for the specific case study. Further, once deciding that it is appropriate, we then need to select and understand the most adequate bias-correction method. In this study, we found ESP to be a 'hard-to-beat' reference not only in terms of skill (as previously found by others, e.g. (Harrigan et al., 2018)) but also in terms of forecast value (Figure 4). In fact, the use of DSP-corr delivers higher energy savings with respect to ESP (without compromising the resource availability) at least in the most relevant operating priority scenario (the *rap* scenario, see Figure 6). However, whether these cost-savings are large enough to justify the use of DSP-corr, or whether water managers may fall back on using simpler ESP, it is difficult to argue and remain an open question with the simulations results available so far.

One point where our results instead point to a univocal and clear conclusion is in the importance of explicitly considering forecast uncertainty (Figure 5). In fact, RTOS outperforms the current operation when using ensemble forecasts, but it does not if uncertainty is removed and the system is optimised against the ensemble mean. In this case, in fact, DSP-corr improves energy savings but it decreases the resource availability under all operational priority scenario. This is in line with previous results obtained using short-term forecasts for flood control (Ficchì et al., 2016), who found that consideration of forecast uncertainty could largely compensate the loss in value caused by forecast errors, hydropower generation (Boucher et al., 2012) and multi-purpose systems (Yao and Georgakakos, 2001). It is also consistent with previous results by Anghileri et al. (2016),

who did not find significant value in seasonal forecasts while using a deterministic optimisation approach (they did not explore the use of ensemble though).

Finally, we tried to investigate whether we could evaluate the effect of the ensemble size on the value of the uncertain forecasts. We found that in our case study we could reduce the number of forecast members down to about 10 (from the original size of 25) with limited impact on the forecast value (Figure 5). This is important for practice because by reducing the number of forecast members one can reduce the computation time of the RTOS. While we cannot say if such 'optimal' ensemble size would apply to other systems too, we would suggest that future studies could look at how the quality of the uncertainty characterisation impacts on the forecast value, and whether a 'minimum representation of uncertainty' exists that ensures the most effective use of forecasts for water resource management.

From the UK water industry perspective, we hope our results will motivate a move away from the deterministic (worst-case scenario) approach that often prevails when using models to support short-term decisions, and a shift towards more explicit consideration of model uncertainties. Such a move would also align with the advocated use of "risk-based" approaches for long-term planning (Hall et al., 2012, Turner et al., 2016, UKWIR, 2016a, UKWIR, 2016b), which have indeed been adopted by water companies in the preparation of their Water Resource Management Plans (SouthernWater, 2018, UnitedUtilities, 2019). The results presented here, and in the above cited studies, suggest that greater consideration of uncertainty and trade-offs would also be beneficial in short-term production planning.

### 4.1 Limitations and perspective for future research and implementation

Our study is subject to a range of limitations that should be kept in mind when evaluating our results. First, the current (and future) skill of seasonal meteorological forecasts varies spatially across the UK depending on the influence of climate teleconnections and particularly the NAO. Given that our case study is located in the South-west of the UK, where the NAO influence has been found to be stronger than in the East (Svensson et al., 2015), our simulated benefits of using DSP seasonal forecasts may be particularly optimistic. Second, the general validity of the results is limited by the relatively short period (2005-2016) that was available for historical simulations, and which may be insufficient to fully characterise the variability of hydrological conditions and hence accurately estimate the system's performances (see for example discussion in Dobson et al. (2019)). Hence, we aim at continuing the evaluation of the RTOS over time as new seasonal forecasts and observations become available. Another limitation is that we used the observed water demand, hence implicitly assuming that operators know in advance the demand values for the entire season with full certainty.

Future studies should extend the testing of the RTOS over a longer time horizon and evaluate the influence of errors in forecasting water demand. To improve our understanding of the forecast skill-value relationship and the benefits of bias correction it would also be interesting to test the sensitivity of our results to the use of different skill scores and bias correction methods. The results of this study and in particular the higher DSP forecast skills than ESP for 1 or 2-month lead times, suggest that combining DSP for the first two months and ESP for the rest of the forecast horizon may be way worthwhile to explore in the future studies.

The Python code developed to: generate the seasonal inflow forecasts from weather forecasts; to optimise the system operation and; to visualise the pre-evaluation Pareto front (with its uncertainty), has been implemented in a set of interactive Jupyter Notebooks, which we have now transferred to the water company in charge of the pumped-storage decisions. The general code and Jupyter Notebooks for application of our methodology to other reservoir systems are available as part of the open-source toolkit iRONS (https://github.com/AndresPenuela/iRONS). This toolkit aims at addressing some of the problems identified in the literature for the implementation of forecast informed reservoir operation systems, by providing better "packaging" (Goulter, 1992) of model results and their uncertainties, enabling the interactive involvement of decision makers (Goulter, 1992) and creating a standard and formal methodology (Labadie, 2004) to support model-informed decisions. Besides supporting the specific decision-making problem faced by the water company involved in this study, through this collaboration we aim at evaluating more broadly how effective our toolkit is to promote knowledge transfer from the research to the professional community and how easily the toolkit can be adapted for different purposes. Through the use of the toolkit, we also hope to gain a better understanding of how decision-makers view forecast uncertainty, of the institutional constraints limiting the use and implementation of this information (Rayner et al., 2005) and of the most effective ways in which forecast uncertainty and simulated system robustness can be represented.

## 5. Conclusions

This work assessed the potential of using a real-time optimization system informed by seasonal forecasts to improve reservoir operation in a UK water supply system. While the specific results are only valid for the studied system, they enable us to draw some more general conclusions. First, we found that the use of seasonal forecasts can improve the efficiency of reservoir operation, but only if the forecast uncertainty is explicitly considered. Uncertainty is characterised here by a forecast ensemble, and we found that the performance improvement is maintained also when the forecast ensemble size is reduced up to a certain limit. Second, while dynamical streamflow predictions (DSP) generated by numerical weather predictions provided the highest value in our case study (under a scenario that prioritise water availability over pumping costs), still ensemble streamflow predictions (ESP), which are more easily derived from observed meteorological conditions in previous years, remain a hard-to-beat reference in terms of both skill and value. Third, the relationship between the forecast skill and its value for decision-making is complex and strongly affected by the decision maker priorities and the hydrological conditions in each specific year. It must be noted that in practice the decision-making priorities are not solely related to the selection of a specific Pareto-optimal solution, but in the first place by the methodology, i.e. the "risk" taken in using something other than the worst-case scenario approach and in applying bias correction of the meteorological forcing or not. We also hope that the study will stimulate further research towards better understanding the skill-value relationship, and in finding ways to extract value from forecasts in support of water resources management.

*Data availability*. The reservoir system data used are property of Wessex Water and as such cannot be shared by the authors. ECMWF data are available under a range of licences, for more information please visit http://www.ecmwf.int. A generic version of the code used for implementing the RTOS methodology is available at https://github.com/AndresPenuela/iRONS.

*Author contributions*. AP developed the model code and performed the simulations under the supervision of FP. CH helped to frame the case study and in the interpretation of the results. All the authors contributed to the writing of the manuscript.

*Competing interests*. We declare that there are no competing interests.

*Acknowledgments*. This work is funded by the Engineering and Physical Sciences Research Council (EPSRC), grant EP/R007330/1. The authors are also very grateful to Wessex Water for the data provided. The authors wish to thank the Copernicus Climate Change and Atmosphere Monitoring Services for providing the seasonal forecasts generated by the ECMWF seasonal forecasting systems (SEAS5). Neither the European Commission nor ECMWF is responsible for any use that may be made of the Copernicus information or data it contains

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

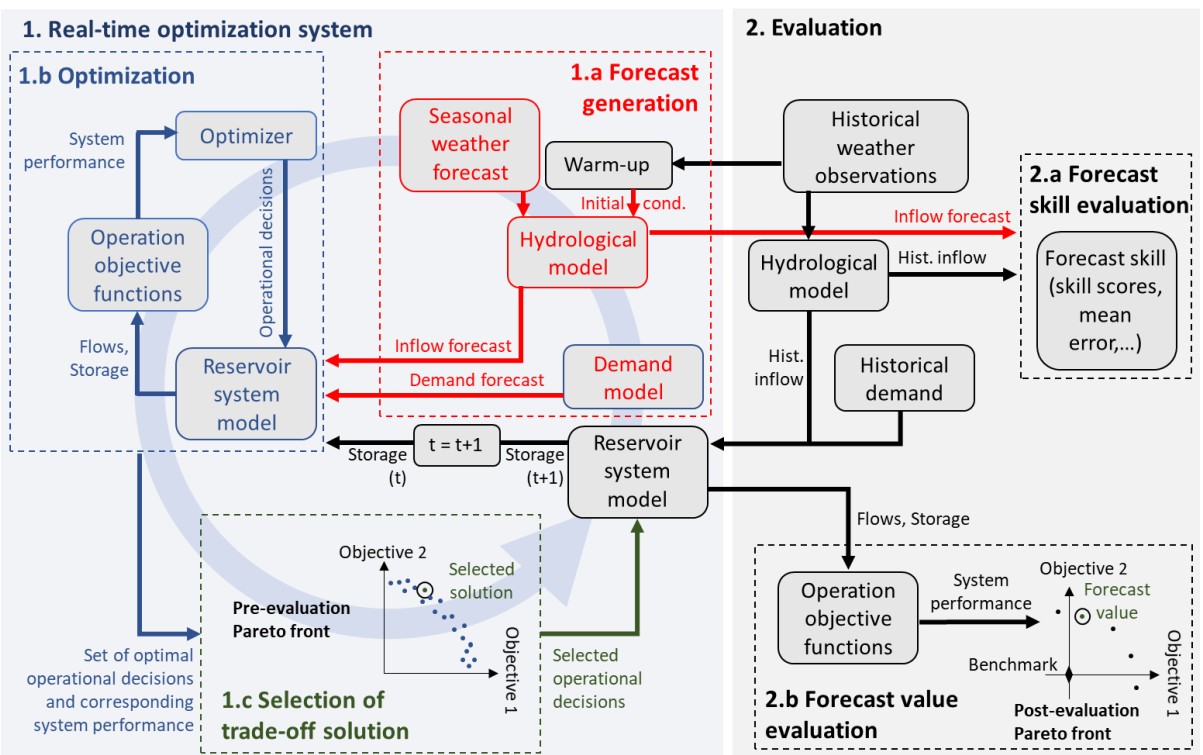

**Figure 1 Diagram of the methodology used in this study to generate operational decisions using a Real-time optimisation system (RTOS) (left) and to evaluate its performances (right). In the evaluation step, the RTOS is nested into a closed loop simulation where at every time step historical data (weather, inflows and demand), along with the operational decisions suggested by the RTOS, are used to move to the next step by updating the initial hydrological conditions and reservoir storage.**

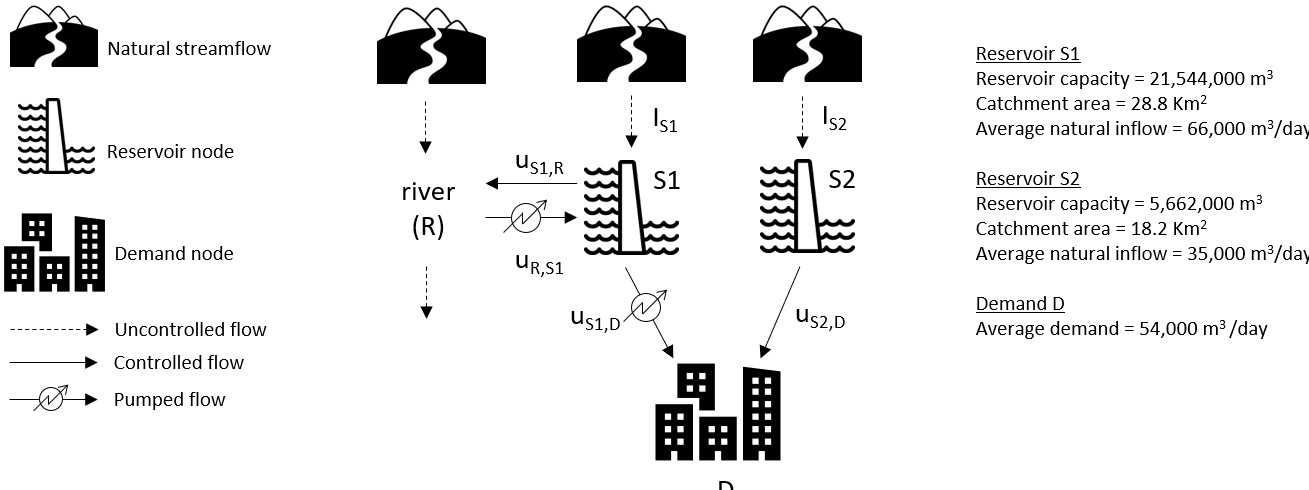

Reservoir S1
Reservoir capacity = 21,544,000 m³
Catchment area = 28.8 Km²
Average natural inflow = 66,000 m³/day

Reservoir S2
Reservoir capacity = 5,662,000 m³
Catchment area = 18.2 Km²
Average natural inflow = 35,000 m³/day

Demand D
Average demand = 54,000 m³ /day

**Figure 2 A schematic of the reservoir system investigated in this study to test the Real-time optimization systems. Reservoir inflows from natural catchments are denoted by I, S1 and S2 are the two reservoir nodes, u denote controlled inflows/releases, R is the river from/to which reservoir S1 can abstract and release, and D is a demand node.**

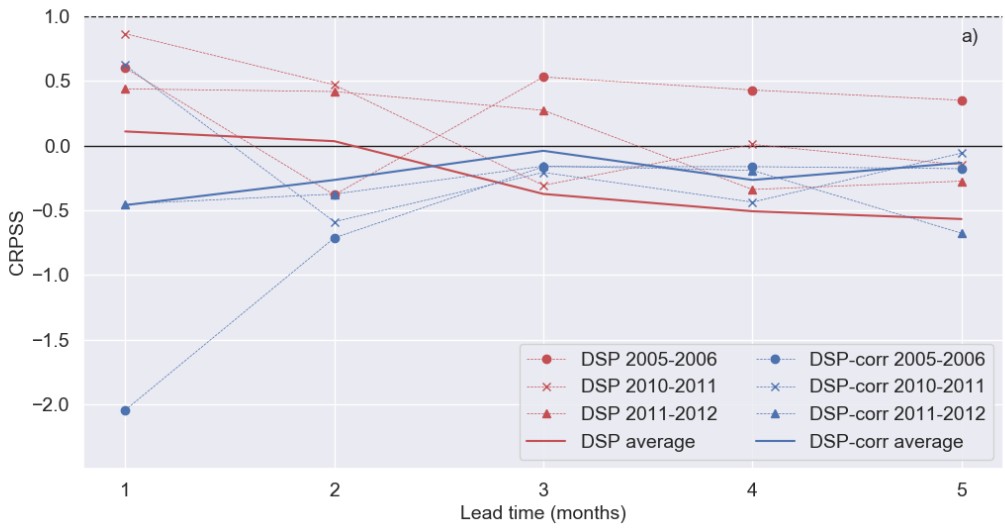

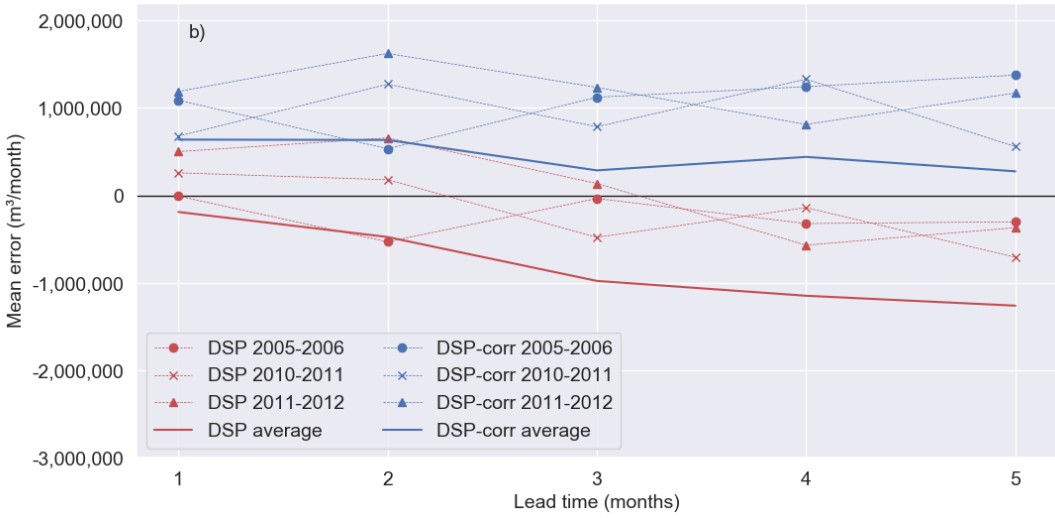

**Figure 3 Skill of the hydrological forecast ensemble (inflow to reservoir S1) during the pumping licence window (1 Nov - 1 Apr) measured by the CRPSS (a) and the mean error (b) for different lead times from 1 Nov. Red lines represent the skill without bias correction of the meteorological forcing (ECMWF seasonal forecasts), blue lines represent the skill after bias correction. Solid lines represents the average skill over the period 2005-2016, while circles, crosses and triangles represent the skill in 3 particularly dry winters (Nov-Apr). CRPSS = 1 represents the perfect forecast and CRPSS = 0 the no skill threshold with respect to the benchmark (ESP).**

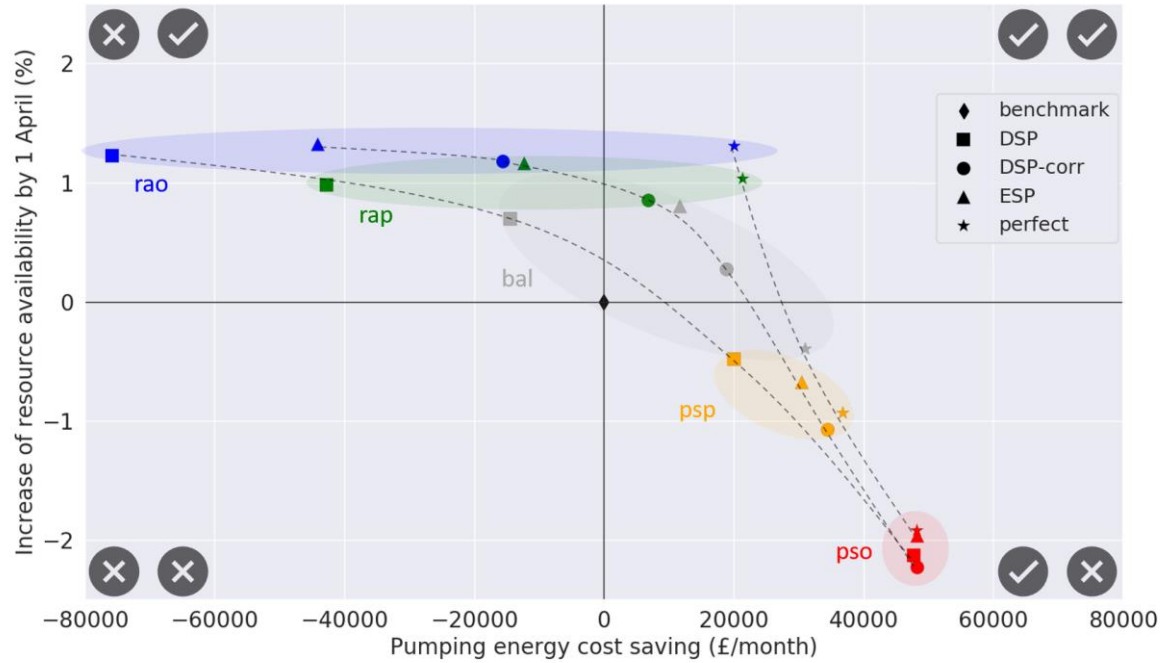

**Figure 4 Post-evaluation Pareto fronts representing the average system performance improvement (over period 2005-2016) of the real-time optimization system during the pumping licence window (1 Nov - 1 Apr) with respect to the benchmark (black diamond), using four forecast products: non-corrected forecast ensemble (DSP), bias corrected forecast ensemble (DSP-corr), ensemble streamflow prediction (ESP) and perfect forecast. For each of the four forecast products, five scenarios of operational priorities are represented: resource availability only (rao; in blue), resource availability prioritised (rap; in green), balanced (bal; in grey), pumping savings prioritised (psp; in green) and pumping savings only (pso; in red). For visualization purposes, the coloured circles group points under the same operational priority scenario and the dashed lines link points using the same forecast product. The pumping energy cost is calculated as the sum of the energy costs associated to pumped inflows and pumped releases and the resource availability as the mean storage volume in both reservoirs (S1 and S2) at the end of the optimisation period. Both objective values are rescaled with respect to the performances of the benchmark operation.**

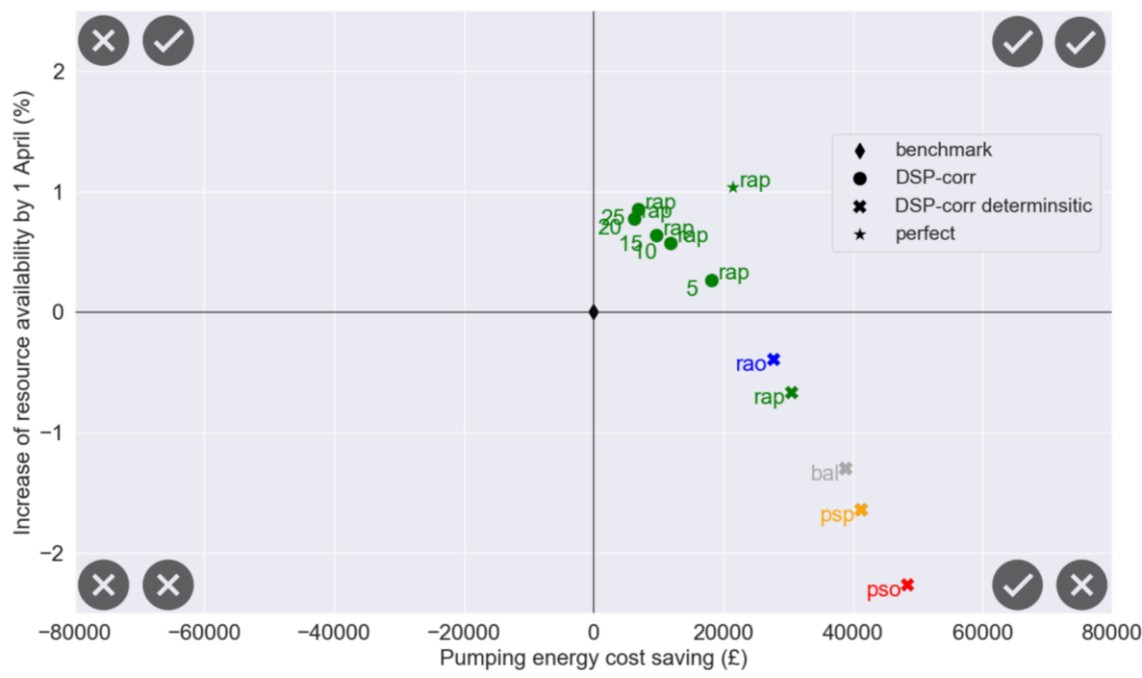

**Figure 5 Post-evaluation Pareto fronts representing the average system performance (over period 2005-2016) of the real-time optimization system during the pumping licence window (1 Nov - 1 Apr) with respect to the benchmark (black diamond), using bias corrected forecast ensemble (DSP-corr) with different ensemble size, and the mean of the forecast ensemble (DSP-corr deterministic). For practical purposes, only the "resource availability prioritised" scenario (rap) is represented for the DSP-corr. The annotation numbers refer to the ensemble size.**

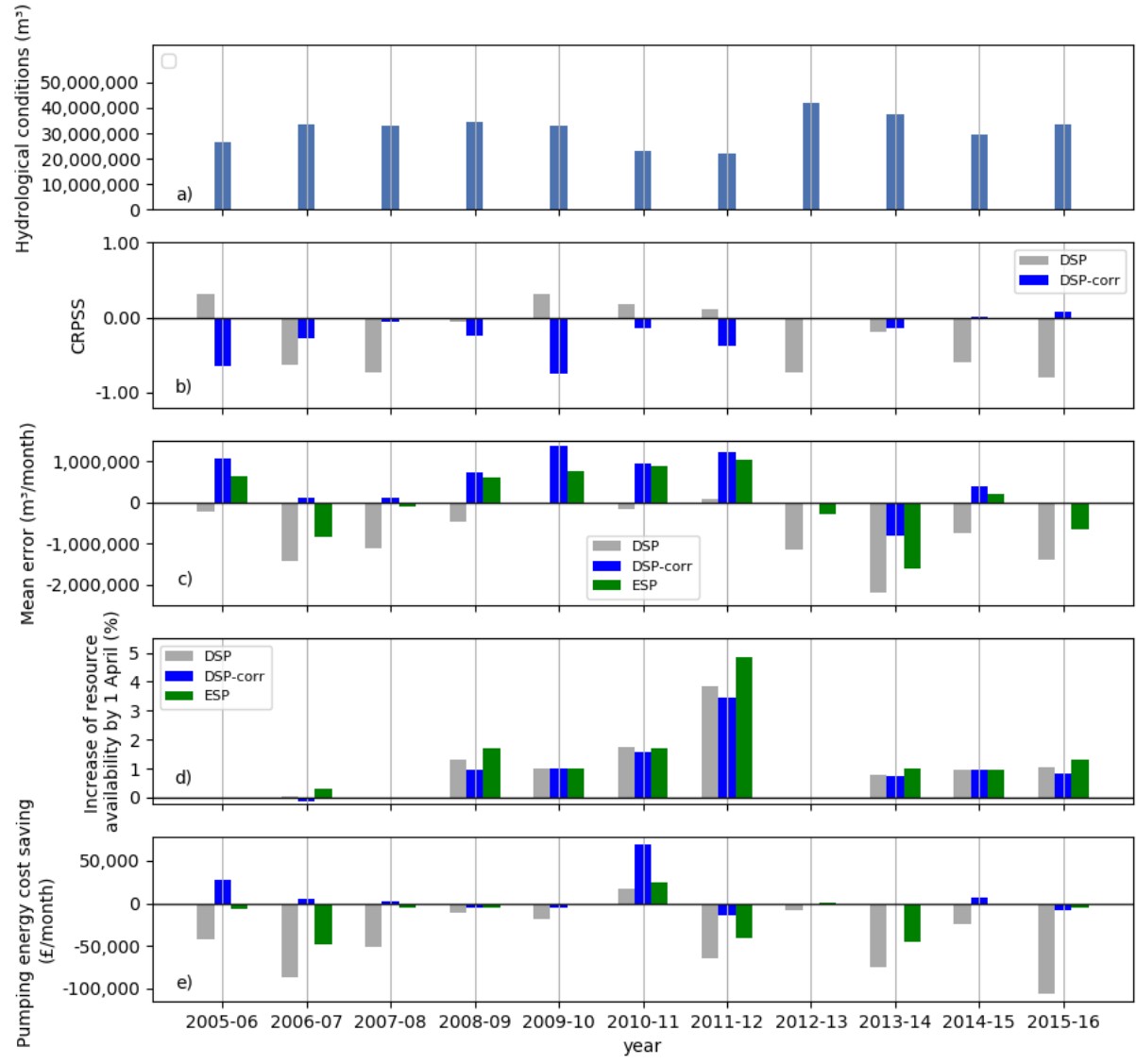

**Figure 6 Year-by-year a) Hydrological conditions (Total observed inflows + Initial storage); forecast skills of the meteorological forcing: b) CRPSS and c) Mean error; d) Increase of resource availability and e) Pumping energy cost savings of the real operation system informed by: the dynamical streamflow prediction (DSP), the bias corrected dynamical streamflow prediction (DSP-corr) and the ensemble streamflow prediction (ESP) for the "resource availability prioritised" (rap) scenario. Please note that ESP is not**

**shown in b) as it is the CRPSS benchmark.**

**Supplementary material**

**Reservoir system model**

We use weekly resolution to simulate the system and its operation for both the benchmark and the real-time optimization system (RTOS) approaches. For each reservoir (S1 and S2), the volume of stored water ($s(t+1)$) is equal to the previous week's storage ($s(t)$) plus natural and controlled inflows minus releases, evaporation and spills. The mass balance equations are:

S1: $s_{t+1} = s_t + (I_{S1,t} + u_{R,S1,t}) - (u_{S1,D,t} + u_{S1,R,t} + evap_t + spill_t + env_t)$

S2: $s_{t+1} = s_t + (I_{S2,t}) - (u_{S2,D,t} + evap_t + spill_t + env_t)$

Spills are calculated by imposing the hard constraint that the storage at next time-step should never exceed the reservoir capacity, hence they are either equal to zero or to the excess volume generated by the storage plus inflows minus outflows:

S1: $spill_t = max(s_t + (I_{S1,t} + u_{R,S1,t}) - (u_{S1,D,t} + u_{S1,R,t} + evap_t + env_t) - s_{max}, 0)$

S2: $spill_t = max(s_t + (I_{S1,t}) - (u_{S2,D,t} + evap_t + env_t) - s_{max}, 0)$

where $s_{max}$ the reservoir storage capacity in m³. Controlled inflows and outflows (u) are limited by the real-world system capacity. Besides, pumped inflows are limited such that flow downstream of R does not drop below a legal constraining value, unless using water released from S1. Evaporation fluxes (evap) are computed as the product of the reservoir surface area by the potential evaporation rate. Environmental compensation flows (env) are given by prescribed values that are kept constant over the year.

**Formulation of the optimization problem**

Both the release scheduling of the benchmark approach and the release and pumped inflow scheduling of the real-time optimization system (RTOS) approach are optimized using the NSGA2 genetic optimization algorithm included in the Platypus Python package (https://platypus.readthedocs.io/). In the RTOS, the optimization decision variables are both the weekly reservoir releases ($u_{S1,D}$ and $u_{S2,D}$) and the weekly pumped inflows ($u_{S1,R}$); in the benchmark operation, the decision variables are the reservoir releases only, while the pumped inflows are calculated according to the control curve. We assume that the future water demand is perfectly known in advance, and equal to the sum of the observed releases from S1 ($u_{S1,D}$) and S2 ($u_{S2,D}$) for the period of study. Unless physically unfeasible, the sum of reservoir releases ($u_{S1,D} + u_{S2,D}$) is always forced to meet such demand.

When simulating the benchmark operation, the optimization is constrained to achieve maximum storage for both reservoirs (S1 and S2) by the end of the pumping license period window (1 April). The (single) optimisation objective is to minimize the sum of the energy costs for pumped release ($u_{S1,D}$) and pumped storage ($u_{R,S1}$):

$$\sum_{t=0}^{T} c_{R,S1} u_{R,S1,t} + \sum_{t=0}^{T} c_{S1,D} u_{S1,D,t}$$

where $c$ is the pumping energy cost and T is the lead time in weeks.

For the RTOS approach, the optimization decision variables are the weekly reservoir releases ($u_{S1,D}$ and $u_{S2,D}$) and the weekly pumped inflows ($u_{S1,R}$) and the optimization objectives to be minimized are two:

1)  Sum of the pumping energy costs (same equation as above)

2)  Average difference between the reservoir capacity and storage volume by 1 April in the two reservoirs (S1 and S2):

$$\frac{1}{M} \sum_{m=0}^{M} \frac{(s_{S1,max} - s_{S1,T,m}) + (s_{S2,max} - s_{S2,T,m})}{2}$$

where $s_{max}$ is the reservoir storage capacity in m³, $s$ is the reservoir storage volume in m³, $T$ is the final week of the optimisation period, and $M$ the total number of ensemble members. Notice that, as denoted by the subscript $m$, the final storage of S1 and S2 will differ depending on the inflow forecast ensemble member that is used to force the simulation, even if the set of pumping and release decisions remain the same. Hence, at each iteration of the optimisation procedure, the same set of decisions is evaluated against each ensemble member and then the objective value is obtained by averaging across all the simulations (with the exception of the "deterministic" case presented in Sec. 3.2.2, where the ensemble forecast is replaced by the mean forecast and therefore averaging is not needed as only one simulation is run against any set of decisions).

For both operation approaches, benchmark and RTOS, the population size for the multi-objective optimization of the RTOS approach was 20.

**Observational hydrological data**

Daily rainfall in the study area from 1981 to 2016 was derived from the UK Centre for Ecology and Hydrology (CEH) Gridded Estimates of Areal Rainfall (CEH-GEAR) dataset (Tanguy et al., 2014) and daily temperature and PET data for the period was derived from the CEH Gridded CEH-CHESS dataset (Robinson et al., 2016, Robinson et al., 2017). CEH-GEAR is a gridded 1km product derived from the interpolation of observed rainfall across all daily and monthly rain gauges in the UK. CEH-CHESS is a gridded 1km product derived from the Met Office 40km gridded MORECS dataset (Hough and Jones, 1997). We used the HBV model forced by observed weather data to simulate a proxy of the daily observed inflows (Figure 7). The HBV model was previously calibrated against observed hydrographs from 1972 to 2003 in the Wimbleball catchment. For the Wimbleball catchment the average observed yearly inflow is 24,462,227 m³/year with an interannual standard deviation equal to 4,340,594 m³/year. Given the lack of good calibration data for the Clatworthy catchment, we applied the Wimbleball parameter set to the Clatworthy catchment, given that they are adjacent to each-other. The averaged mean error from 1972 to 2004 of the Wimbleball calibrated inflow is 33,283 m³/month (from 1 Nov to 1 Apr).

 **Supplementary figures**

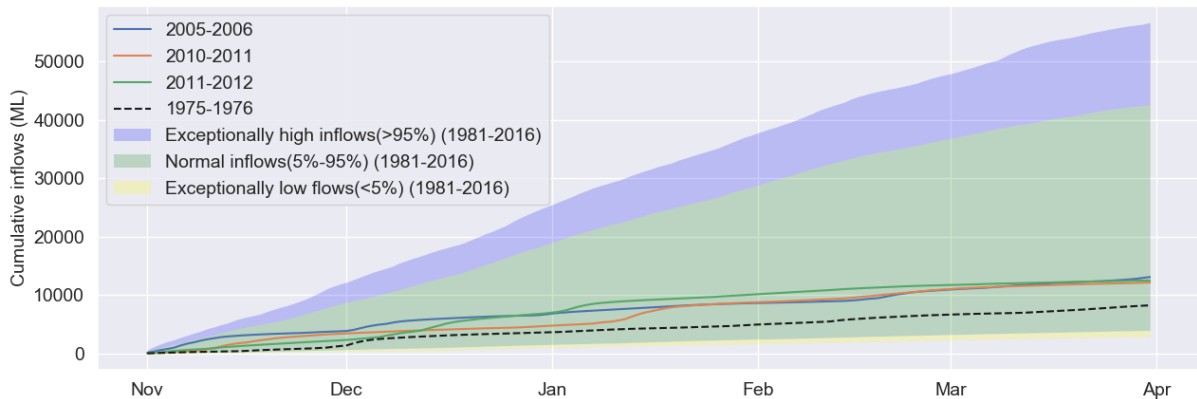

**Figure 7 Cumulative inflows to the S1 reservoir in the worst-case scenario (1975-1976) and in the three driest years (2005-2006, 2010-2011 and 2011-2012) of the period used for the simulation of the RTOS (2005-2016). Only data relative to the pumping licence window (Nov to Apr) are shown. Shaded areas show the weekly inflow distribution calculated on the period used for the forecast bias correction of the meteorological forcing and ESP generation (1981-2016). Notice that the three driest years are relatively close to the worst-case scenario (1975-1976).**