# Peer review of "Assessing the value of seasonal hydrological forecasts for improving water resource management: insights from a pilot application in the UK"

_Hydrology and Earth System Sciences, 2020_

## Referee Comment (RC1) · Anonymous Referee #1 · 6 Apr 2020

This an interesting and timely study into the value of forecasts for improving the performance of a simple water supply reservoir system with an operational trade off between augmentation of stored water through pumping and associated energy cost. The selected case study is appropriately simple and also informative for this type of analysis. Results are quite difficult to follow and key details are omitted from the method. The set of forecasts selected for use in the simulation are also poorly justified. Finally, I feel that the paper attempts to answer too many questions and would benefit significantly from more focus. For example, the analysis of the dynamical forecast product and its failure to provide skill over ESP is an interesting study in its own right, demanding much more in-depth analysis and interpretation than is offered in the paper. The operational

section then addresses ESP vs dynamical and the additional question relating to importance of incorporating ensemble uncertainty. The paper would be much stronger if you were to focus on just one of these areas and deliver a more compelling conclusion backed up with in-depth analysis of a specific question. I recommend that the paper would be publishable if significant changes are made to simplify the overall story and provide further method detail as outlined in the comments below.

Specific comments:

- It's not clear what optimization framework is used to deal with the forecast ensemble. The deterministic approach using rolling horizon (e.g., ANGHILERI et al., 2016) is quite common and there are very few successful examples in the literature where the full ensemble is used to inform the decision. Please outline exactly how the ensemble is used in your optimization and then justify the approach. If this is a new approach it perhaps needs to be described in its own, separate publication.

- Given the skill scores achieved for the dynamical forecasts, it's not clear why these were pursued in the operational part of the study. What is the justification for using a forecast product that is demonstrated to be unskillful relative to ESP?

- I found the results quite difficult to follow, partly because it's hard to keep track of the various operational settings. Why not simplify by showing the Pareto front for each forecast set (as opposed to five schemes with different symbols/colors). This would be both more comprehensive and easier to understand. Also, the emojis in the key figures are not appropriate.
* * *

---

## Referee Comment (RC2) · Anonymous Referee #2 · 6 Apr 2020

**Review of**

**Assessing the value of seasonal hydrological forecasts for improving water resources management: insights from a pilot application in the UK**

**Peñuela, Hutton and Pianosi**

The main merit of this paper is the proposal of the methodology. The paper forms a valuable contribution to the methodology of quantifying the value of forecasts, here in terms of water availability and energy cost. Probably the methodology is more widely applicable. Generally, the paper is well written, although sentences tend to be too long and their structure could sometimes be made clearer by repeating some short words.

Unfortunately, the conclusions from this paper are not really valuable. The problem with the first two conclusions, namely 1) seasonal forecasts can increase value and 2) ESP is hard to beat, is that they are case specific, as acknowledged by the authors (line 470). The third conclusion (the relationship between forecast skill and value is complex) is a trivial one.

Below, there is a quite long list of main points, which the authors have to address in my opinion: information about the observations should be given (p1), any procedure based a scenario or forecasts with more inflow than in the worst-case scenario seems beneficial, e.g. taking the median of the historical years (p2), the methodology should be better explained (ps 3, 5 and 6), Mliters are not a valid unit (p4), there is an issue with the bias correction (p7), different processing for the benchmark and other forecasts is questionable (p8), it is strange that the value of the driest years with DSP and ESP processing is not almost equal to the benchmark (p10) and the first part of the discussion section should perhaps be removed (p9).

In my opinion should be published after making the suggested major revisions.

**Main points**

1) A section about observations (discharge and meteorological forcing) should be added.

2) One of the results of this paper is that by basing the operational procedure on the forecasts, less energy for pumping is used while ensuring similar water availability (statistically over the years), compared to basing operational procedures on the worst-case scenario (driest historical year). It is my impression that any operational procedure based on forecasts or scenarios with more inflow into the reservoirs than in the worst-case scenario leads to less pumping and similar water storage, provided the increases are realistic. The authors confirm this in lines 395-397 for the case of applying a bias correction, which increases the inflows to the reservoirs and hence increases the value of the forecasts. So, the worst-case scenario is possibly easy-to-beat by any scenario with more inflow. Somewhere in the paper (in the discussion section?) the following points need to be discussed. Can this effect on value of increasing the inflow be generalized? What is the value of the forecasts if the operators base their procedure on the scenario of the year with the median value of the historical inflows? I suggest making a calculation with such a scenario. By the way: are the calculations in the worst-case scenario deterministic?

3) I did not understand Section 2.1 – 1b and c. These paragraphs need to be rewritten. At this stage this paragraph is too abstract. Perhaps providing an example of each concept (operation objective function, optimizer, set of operational decisions) would help. Perhaps merging

Sections 2.1 and 2.3 helps. Moreover, after 1b the "set of operational decisions is determined", so why is the operator again "selecting a set of optimal decisions" in 1c? Perhaps lines 124-126 helped me to understand a little bit of what you try to explain, namely that you use hindcasts to evaluate the performance of RTOS. If this is correct, just write that you use hindcasts to evaluate RTOS and discuss a possible operational application in the discussion section.

4) Replace all appearances of Ml and Ml per some time unit by $m^3/s$ (per day is also ok), the common unit in the hydrological literature, e.g. in Figure 2, 3, 6 and 7.

5) I did not completely understand 2.3.1. Was river R fed by the outflow of reservoir S1 before the dam of S1 was built? Also, it sounds ridiculous to pump water, that was released by gravity from S1 to R, back to S1. So, is the water in R at the location where it is pumped out of the river partly fed by rivers that are not connected to S1? Is S1 located at lower elevation than D, so the water flow needs to be pumped?

6) I did not really understand those "rule curves" (lines 173-179). Add a figure with a rule curve. It is not clear to me how the refilling ($U_{R,S1}$) is done. Is the "missing water" immediately refilled or is the refilling spread over time until April 1, using the optimizer? In the latter case, how does the optimizer work? Can you give an example of an operational decision? What level is targeted on April 1? How does the operational procedure work for probabilistic forecasts? Since there is variation in resource availability by April 1 (e.g. in Figure 4), storage is not equal to the target on April 1. Can storage be larger than the target or is all water above the target spilled? Can storage be less than the target? Perhaps only in S2 and not in S1 because water can be pumped into the latter basin?

7) The method of bias correction is not correct (199-203). The number of years used to compute the multiplication factors differ per target year. I suggest using the common leave-one-year-out-method, i.e. the factor for each target year is computed from the data of all other years, including years later than the target year. Your method suggests that it is not allowed to use data from future years but there is no problem in doing so if different years are independent of each other.

8) Line 264 "but with three main variations": Why do you treat the benchmark differently? This implies that if the forecast is equal to the benchmark, the forecast value differs, which seems undesired.

9) The first general lesson in the discussion is "First, we found that the use of bias correction to improve the skill and value of DSP forecast is less straightforward than possibly expected" (lines 381-382). I do not agree. Such an expectation, namely that the forecast skill generally improves due to bias correction (is that the expectation?), just does not exist. Your study indeed confirms that this is a naive expectation. So, remove lines 378-393 or reformulate them. By the way: your bias corrections are based on observations of precipitation and temperature and not on the output (hydrological variables!) of ESP forecasts. So, I did not understand the sentences related to ESP in lines 387-388 and 391-392.

10) It is strange that the increase in the value of the system with DSP or ESP forecasts relative to the value of the system based on the worst-case scenario is highest in the driest years (e.g. lines 408-409), while those driest years resemble the worst-case scenario more than the other years. You need to explain this.

**General text points**

1) The authors often use the term "bias correction" without mentioning what is corrected. As far as I understood, the forcing of the hydrological model is corrected but if you do not repeat mentioning this now and then, it is confusing because the output of the hydrological model, i.e. inflow to the reservoirs, can also be bias-corrected. So, replace at numerous places "bias correction" by "bias correction of the meteorological forcing" or "bias correction of the forcing".

2) In general sentences are too long, making the manuscript difficult to read. So, shorten sentences where there is an opportunity and make the structure of long sentences clearer, especially by adding some words in sentences with "and". I made some suggestions below (e.g. 15-16 in the abstract).

3) "Uncertainty (considerations)" is used to discuss effects of ensemble size and the probabilistic nature of the forecasts. Replace throughout the paper the vague term "uncertainty (considerations)" by the more explicit terms "ensemble size" and "probabilistic/deterministic nature of the forecasts".

**Minor points**

16       Insert "to" before "other factors".

17       "Some of these factors" is too vague. Write which factors have a significant correlation with forecast value (see point Figure 6 below).

24       Add reference to endorse the statement that climate variability is increasing.

44-45    Replace "it provides" by "they provide" and add "that they" before "reflect".

54       I miss the logic behind "i.e. ESP …". Replace this part of the sentence or clarify the logic.

55-56    Reorder sentence to "The possible improvements of supply-hydropower systems operation due to the use of ESP were assessed by Alemu …"

69-71    Remove this sentence: this distracts too much.

78       weaker compared to ….?

83-92    Remove these sentences about some of the many existing metrics. It is not efficient to read about metrics not used in this paper and the metrics used in this evaluation are introduced 2.3.5.

94       Replace "this" by "the".

99       Replace "simulate and compare" by "assess". Simulate performance sounds strange and it is not clear from the rest of the sentence what is compared with what.

151      Insert "diagram" after "Pareto front".

163      Consider removing all text about two companies. It is irrelevant for your story while it is making your story more complex.

168      Insert "(R)" after "river"

188      Remove period.

| | |
|---|---|
| 198 | Did you also use a multiplicative factor for temperature? |
| 211 | $U_{S1,D}$ is also a pumped water flow according to Fig. 2. |
| 214 | Replace "The first objective function" by "Pumping savings" and "The second function" by "Resource availability". |
| 219 | Replace "the 15%" by "only 15%". |
| 227 | Rephrase sentence as follows: "They represent five different trade-offs of operational priorities, according to their relative importance" |
| 237 | Remove sentence. |
| 247 | Remove "and for a given lead time". The role of lead time comes some sentences below. |
| 251 | Replace "lead time" by "range of the lead time (we use monthly ranges)" and "CRPS values" by "individual CRPS values". |
| 252 | Replace "CRPS" by "individual CRPS values". |
| 253 | I suggest to replace "mean error" by "discharge bias" since bias in the common word for mean error and the addition of "discharge" helps to distinguish this bias from that in the forcing. I also find the equation redundant. Just write that the bias is the difference between the means of the forecasts and the observation over all ......." |
| 262 | Add "(1975-76)" after "drought on records". |
| Figure 3 | Is this the sum of the inflows to both reservoirs? Are these results for the whole year or a specific part of the year? In the legend of the lower panel "2006" should be replaced by "2016". |
| 334 and 338 | Replace "Figure 6" by "Figure 5". |
| Figure 6 | If I just look at these graphs, I get the impression that there is no significant relationship in any of these graphs. However, according to your p-values relationships are significant at the 90% confidence level in panels b and e. Is the calculation of the p-values correct? Or are those low p-values due to using the Spearman coefficient instead of the Pearson coefficient? I think you should use Pearson unless you have good reasons to use Spearman. |
| 344 | Replace "skill" now and then by "forecast skill", to remember the reader what type of skill this is. |
| 358 | Replace "year" by "years". |
| 395 | Replace "reduce" by "reduces". |
| 399 | "improvement of forecast accuracy in some direction". What do you mean by "accuracy"? For me this is something like the root-mean-square-error, which means that there is only one desired direction, namely towards 0. Do you mean something like "a change towards either higher or lower values can be more valuable than a change in the other direction"? |

411      Is "Initial storage (total storage value)"equal to "Initial storage" in Figure 6? For clarity, be consequent in the use of specific terms. Moreover, panels c and h in Figure 6 do not show a significant correlation (p-values of 0.21 and 0.80).

428      Remove ")"

447      Replace, for clarity, "seasonal forecasts" by "seasonal meteorological forecasts".

465      Insert "of" before "the institutional"

466      Insert "of" before "the most"

478-479 Replace "but also the methodology in the first place" by "but in the first place by the methodology".

---

## Referee Comment (RC3) · Anonymous Referee #3 · 6 Apr 2020

Review of "Assessing the value of seasonal hydrological forecasts for improving water resource management: insights from a pilot application in the UK" by Andres Peñuela, Christopher Hutton and Francesca Pianosi

This paper addresses comprehensively a topic that increasingly requires investigations: the value of seasonal forecasts for real-life applications. Until recently, many studies have worked on assessing or improving forecast skill, but still few manage to link this skill to value. In addition to being innovative, this study investigates the issue through different uncertainty lenses which makes it a very strong contribution to the field and results in valuable findings both for the scientific community but also for water management stakeholders. Overall, the paper, its ideas, structure and methodology are of high quality. However, additional information, for example about the data used and the forecast skill evaluation, would be necessary to support the analysis. In addition, I have some concerns about the validity of the results on linking skill and value due to the chosen methodology. These are detailed hereafter.

**General comments**

- The paper would benefit from a Data section, presenting for instance the data used as reference in the bias correction, the inflows used as reference in the forecast evaluation, or the demand model/observations.
- Some additional information would be needed in Section 2.2 on the forecast skill evaluation. More specifically, (1) In Figure 2, two inflows feed the two reservoirs (Is1, Is2), which inflow is being considered when evaluating forecasts? (2) Which time period is used to evaluate the forecasts? In Figure 8, November to April is shown, but in the forecast methodology (Sections 2.2 and 2.3.2) or in Figure 3, no specific time period is mentioned. I would recommend mentioning these two points in 2.1/2a.
- Since the goal of the paper is to assess the added value of dynamical seasonal forecasts for water management, and since authors evaluate the skill and performance of these forecasts against observations, it would be important: (1) to add information about how HBV was calibrated and setup for the area, or at least, to mention its performance (mean error) in simulating past inflows to the reservoirs (giving the possibility to make a parallel with the results in Figure 3); (2) to mention even briefly the hydrological regime upstream the reservoir system, as well as the interannual variability, which will define the added value of a method like DSP over ESP.
- I have concerns about the methodology chosen to link value and skill, and therefore about some of the subsequent results (first two paragraphs of Section 3.2.3). The authors are trying to link a skill obtained from comparing dynamic forecasts with forecasts based on past climatology (1981-20XX), with a value obtained from comparing dynamic forecasts with a worst-case scenario (1975/1976). On the one hand, the skill is based on the comparison of DSP-corr with ESP, meaning that its value solely indicates the gain in performance from using ECMWF SEAS5 instead of historical precipitation and temperature. On the other hand, the value is obtained by comparing DSP-corr with a benchmark based on a worst case scenario (1975-1976). These choices result in skills and values that cannot be compared. Instead, the skill computed in the paper could be related to the gains/losses in value between DSP-corr

and ESP (difference between the blue and green bars of Figure 7). Reversely, the value computed in the paper could be compared to a skill whose benchmark would be the performance obtained by using the worst case scenario (1975-76) as forecast in all years. This is a major point that should be addressed prior to publication to ensure the validity of the conclusions.

**Specific comments**

L45: "and reflects the risk-averse attitude"

L68: "to continue increasing in coming years"

L75-76: More literature review would be needed on past works on seasonal (climate or hydrological) forecasts value. Here are a few examples to consider:

Bruno Soares, Marta, Meaghan Daly, and Suraje Dessai. 'Assessing the Value of Seasonal Climate Forecasts for Decision-Making'. *WIREs Climate Change* 9, no. 4 (2018): e523. doi:10.1002/wcc.523.

Giuliani, M., L. Crochemore, I. Pechlivanidis, and A. Castelletti. 'From Skill to Value: Isolating the Influence of End-User Behaviour on Seasonal Forecast Assessment'. Hydrology and Earth System Sciences Discussions 2020 (2020): 1–20. doi:10.5194/hess-2019-659.

Parton, Kevin A., Jason Crean, and Peter Hayman. 'The Value of Seasonal Climate Forecasts for Australian Agriculture'. *Agricultural Systems* 174 (2019): 1–10. doi:10.1016/j.agsy.2019.04.005.

L107: In this sentence, the authors seem to assume that their results could be generalized to extra-tropical regions and that the potential for use is region-dependent. However, as the authors state, results (forecast value) are in fact highly dependent on the investigated system, and therefore this sentence may sound a bit too ambitious compared to the paper objectives.

Section 2.1: I found this section very clear and helpful.

L118: Here, I was wondering what type of demand model was used. It is later in the discussion that the authors mentioned that the demand is based on observations. This would be worth mentioning earlier in the text, and would fit in a Data section.

L142 (2.a): More information would be needed at this stage about the forecast skill evaluation, for example by mentioning the reference and the benchmark used.

L160-161: The notation for units is not consistent between the text (Ml) and Figure 2 (ML).

L176: Did you mean a "reasonable chance"?

L182: Please consider adding the following reference for ECMWF SEAS5:

Johnson, S. J., Stockdale, T. N., Ferranti, L., Balmaseda, M. A., Molteni, F., Magnusson, L., Tietsche, S., Decremer, D., Weisheimer, A., Balsamo, G., Keeley, S. P. E., Mogensen, K., Zuo, H. and Monge-Sanz, B. M.: SEAS5: the new ECMWF seasonal forecast system, Geoscientific Model Development, 12(3), 1087–1117, doi:10.5194/gmd-12-1087-2019, 2019.

L182-183: I suggest writing "The ECMWF SEAS5 hindcast dataset" because it is the hindcast dataset that includes 25 members, while the real-time operational ECMWF SEAS5 has more members.

L187: There are two points on this line.

L195: This statement needs to be refined. Linear scaling and distribution mapping may lead to similar results in terms of bias removal, but they will have very different results on the forecast themselves, e.g. linear scaling will just shift the ensemble, while distribution mapping will have a different correction for each member, resulting in larger impacts on the spread.

L196-198: Usually a difference approach is applied to calculate the correction factor for temperatures (see e.g. Lucatero et al. 2018, Teutschbein and Seibert 2012). Similarly the correction factor is added/subtracted to correct the raw forecasts. If a ratio approach was applied for temperatures, I would suggest adding a short justification of this choice. For instance, how are negative temperatures handled?

Lucatero, D., Madsen, H., Refsgaard, J. C., Kidmose, J. and Jensen, K. H.: On the skill of raw and post-processed ensemble seasonal meteorological forecasts in Denmark, Hydrology and Earth System Sciences, 22(12), 6591–6609, doi:10.5194/hess-22-6591-2018, 2018.

Teutschbein, C. and Seibert, J.: Bias correction of regional climate model simulations for hydrological climate-change impact studies: Review and evaluation of different methods, Journal of Hydrology, 456–457, 12–29, doi:10.1016/j.jhydrol.2012.05.052, 2012.

L206: "with what was done"

L208: The ensemble size has an impact on the CRPS (Ferro et al. 2008) and therefore, on the skill when the benchmark has a different ensemble size than the evaluated system. This could impact the results presented in this paper, and I would suggest adding comment about the impact of the varying ESP ensemble size on your skill results.

Ferro, C. A. T., Richardson, D. S. and Weigel, A. P.: On the effect of ensemble size on the discrete and continuous ranked probability scores, Met. Apps, 15(1), 19–24, doi:10.1002/met.45, 2008.

Section 2.3.5: I would suggest presenting the CRPS (L245-252) before presenting the CRPSS (L239-244) since it would define the CRPS notation before using it in the skill equation. There are also inconsistencies between the first two equations (CRPS and CRPSS) and the third one (mean error), such as the notation for the observations ($y$ vs $I^{Obs}$), the notation for the system (*Sys* vs *Syst*), the mention of the lead time only in the third equation but not in the previous ones.

L237-238: This sentence is repeated twice.

L243: By choosing ESP as a benchmark, you also make the decision of only analysing the added skill from using dynamic weather forecasts over meteorological history, it would be worth mentioning.

L256: Do I understand correctly that the mean error is computed over a time window that varies with the lead time you consider, for example to evaluate lead time 6 weeks, you average the mean error from week 0, 1, … to 6? Is the same calculation method applied when computing the CRPS?

L279: I suggest replacing "gets lower" with "gets larger in absolute value".

L334: The reference should be to Figure 5 instead of Figure 6.

L358: "two specific years"

L381-393: In this paragraph, conclusions are likely only valid for linear scaling, and not for any bias correction. I would suggest being more specific throughout the paragraph.

L446: "that should be kept in mind"

Figure 6: I wonder why, for each of the five operation scenarios, the authors do not display both DSP-corr and ESP in the same graphs.

Figure 6b and 6g: Shouldn't the absolute mean error be used to assess the correlation between the increase of resource availability and the performance, since both high positive and low negatives reflect poor performance?

Figure 7: It could be interesting to see the mean error and the CRPSS in this figure, even if they are not correlated.

Supplementary material – Figure 8: In the legend, I could not remember that 1975-1976 was the worst-case scenario. It could be worth mentioning it again at the end of the caption.

Supplementary material – Figures 9 to 13: The CRPSS is used to try and explain the gains in value from using ESP. Here, it is not clear which benchmark the performance of ESP was compared to in order to compute the skill, if it is indeed the CRPSS of ESP being shown. If not, then please refer to the fourth general comment.

---

## Editor Comment (EC1) · Nunzio Romano (Editor) · 7 Apr 2020

Dear Authors,

With a view to further stimulating this interesting discussion phase on your contribution and receiving possible additional reactions from the discussants, I would suggest you should start providing some preliminary responses to the comments received so far. This can also help organize more effectively your final responses when required after the deadline of the D-phase (in early May).
* * *
89, 2020.

---

## Referee Comment (RC4) · Anonymous Referee #4 · 27 Apr 2020

Dear authors,

Thank you for this interesting research, written up in a well-organised and clear paper. Overall, I have not hesitation to recommend your paper for publication. I do agree with you, that this kind of research, with continuous simulation of operational water management to test new information sources, methods, or strategies, is valuable for science and in particular for bringing findings forward to practice in an informed and iterative way.

I appreciate in particular your Conclusions section, and clear description of data used, methodology, and presentation of results.

[Figure]

General comments

I have the following main comments:

- The authors assumed the water demand to be known in advance and to be equal to the observed reservoir releases (line 220). This may be an important assumption. If more than needed was pumped-in for storage, this would perhaps also lead to releasing more than the actual demand. Could the authors reflect on this? The actual releases are the result of the current water management priorities, which focus on water resources availability. Could this have led to the forecast value also being maximum for the rap scenario's? Could the authors address this with a limited sensitivity analysis, varying water demand? (if time, sensitivity analysis of other aspects would be interesting as well, e.g. towards the set-up and settings of the NSGA optimisation experiment.)

- To my view, the results show that the bias correction applied, did not work in this particular case study, Figure 3 (only changed sign of MAE from under- to over-prediction, as authors also indicate in line 364). Could the authors reflect on this in their section on limitations of the research? What may be the reason? Could other bias correction methods work better or is this not to be expected (e.g. perhaps higher forecast skill is needed to begin with, for post-processing methods to be effective)? The poor performance of the bias correction also connects to the following comment.

- The Discussion section contains notes and even recommendations on the use of bias correction. In my view, the poor performance of the bias correction in this case study, and the fact that, as the authors point out, indeed there is only one particular case study analysed here, do not warrant such discussion on the merits of bias correction. Could the authors reflect on this, and depending on whether they agree or disagree, adjust the Discussion accordingly.

- The Discussion section also recommends use of ensemble (probabilistic) forecasts in operational management, which is supported by the research findings, but then connects this to UK policy recommendations on long-term water resources planning. This I think is a bridge too far, and not needed. I would favour the Discussion to be less broad, and stay focused on the research findings presented (see my detailed comments for specific suggestions on where and how to make the Discussion section more specific). This leads to my next comment.

- I miss a more in-depth discussion on the forecast skill of the DSP used, and the influence of forecast lead time throughout the analysis chain. The CRPSS results nicely show that only for the first two months the uncorrected forecasts have skill (the bias-forecasts do not have skill). Is this positive skill utilised by the operational water management strategies simulated. Could the authors suggest ways on how to capitalise more on this positive skill, e.g. by using DSP for the first 2-month lead time, and using ESP for months 3 to 6?

- Lastly, to come back to the motivation of the authors to bridge science to practice, I would like to see observed and simulated releases for sample priority scenarios and years. These actual releases throughout a season is what operators will recognise and this will enable a discussion on how and to what extend the use of ensemble seasonal forecasts would lead to changes in operation.

Detailed comments:

- line 275: Indeed the bias corrected DSP "improves" skill for longer lead times, but only from negative skill to less negative skill. I would suggest to point that out.

- line 281: Yes, with bias correction there is "some improvements for some lead times", but still with negative skill. Rather than pointing out "some improvements for some lead times", I think it is more relevant to point out here that the bias correction as applied here, in this case study, is not working and even has adverse effect on forecast performance.

- line 311: The question is why? Again (See my first general comment, and note that resource availability in the results varies only with 1-2%) this may indicate a constraint set-up, favouring a focus on rao. Please reflect on this.

- line 382: "Our results show that on average bias correction slightly improves the DSP forecast skill (as measured by CRPSS and mean error)". I do not agree here. When looking at the results, also on average, bias correction reduced the CRPSS for the first 2 months lead time where the DSP had skill (bias correction made skill less negative for further lead times, but still negative), and it changed the sign of the average error but did not reduce it, so bias correction was not working well.

- line 392: I agree with the authors. Based on this particular study, not much can be discussed on the merits of bias correction. Instead I would recommend to focus discussion in the paper more on the skill of the ensemble DSP (slightly positive for the first 2 months), how and why this is or is not being used in the water system operations simulated (see my last two General comments above).

- lines 399-404: While I agree with this statement in general, I do not think the research presented provides sufficient supporting evidence. Nor is it a surprise or a new finding.

- lines 413-416: This is quite a strong recommendation not substantiated with any analysis/numbers on what these 'costs' are. The authors could consider leaving this out.

- lines 433-436: I am not sure if the link with Long-term water resources planning is appropriate, and also I think it is not needed to make the case of risk-based operation on the basis of ensemble (probabilistic) forecasts. The results of your study do show this nice enough. The authors could consider leaving this out.

- lines 458-467: Yes, developing such toolkit and making available to the water management organisation is very valuable. Indeed question here would be to what extend the toolkit is customised to the specific case study, and how much time/effort customisation to a new case study would require. Could the authors reflect on this in the text?

- line 475: DSP is now more readily available from international weather forecast centres and more easily processed, such that this by-sentence on "ESP being more easily derived", in my view is perhaps becoming less relevant. Also because, as the authors describe, they have provided a Toolkit for ease-of-use.

Please see for suggested technical changes (editorials) the annotated pdf.

Thank you and with best regards.

Please also note the supplement to this comment:
https://www.hydrol-earth-syst-sci-discuss.net/hess-2020-89/hess-2020-89-RC4-supplement.pdf

**Supplement:**

[revised manuscript text omitted]

---

## Author Comment (AC4) · 11 May 2020

**Throughout this response, the reviewer's text is presented in black, our response in blue**

Dear authors,
Thank you for this interesting research, written up in a well-organised and clear paper. Overall, I have not hesitation to recommend your paper for publication. I do agree with you, that this kind of research, with continuous simulation of operational water management to test new information sources, methods, or strategies, is valuable for science and in particular for bringing findings forward to practice in an informed and iterative way.
I appreciate in particular your Conclusions section, and clear description of data used, methodology, and presentation of results.

We thank the reviewer for their kind words.

General comments
I have the following main comments:

- The authors assumed the water demand to be known in advance and to be equal to the observed reservoir releases (line 220). This may be an important assumption. If more than needed was pumped-in for storage, this would perhaps also lead to releasing more than the actual demand. Could the authors reflect on this? The actual releases are the result of the current water management priorities, which focus on water resources availability. Could this have led to the forecast value also being maximum for the rap scenario's? Could the authors address this with a limited sensitivity analysis, varying water demand? (if time, sensitivity analysis of other aspects would be interesting as well, e.g. towards the set-up and settings of the NSGA optimisation experiment.)

The release data that we use are only the controlled releases (from the outlet tower) and do not include spills, so we believe that our assumption that they reflect the demand is quite ok. Moreover, in the system, we have only modelled the refill period during winter where demands on the system are fairly stable/predictable. The forecast value is quantified with respect to the benchmark and both benchmark and DSP under all the scenarios assume the same demand. So very unlikely that changing the demand is going to have an influence on how better DSP does with respect to the benchmark. The point made about not knowing demand in advance is more relevant if this case were expanded to understand what happens with demand in the following summer, and whether we would really need to refill the reservoir or not. We will clarify these points in the revised manuscript. A sensitivity analysis is a good idea that we would like to address in future publications, but this is out of the scope of this study and besides, reviewers complained about this paper being too long. The message that we want to convey with this study is that yes, we can improve the performance of a realistic reservoir system using seasonal forecast using common and readily available methods and forecast products. Nevertheless, in the revised manuscript we will further discuss the possible reasons that have led to the forecast value being maximum for the rap scenario, such as the one pointed by the reviewer.

- To my view, the results show that the bias correction applied, did not work in this particular case study, Figure 3 (only changed sign of MAE from under- to over-prediction, as authors also indicate in line 364). Could the authors reflect on this in their section on limitations of the research? What may be the reason? Could other bias correction methods work better or is this not to be expected (e.g. perhaps higher forecast skill is needed to begin with, for post-processing methods to be effective)? The poor performance of the bias correction also connects to the following comment.

The main reason for the bias correction to fail is the DSP forecast was already doing relatively good in terms of skills in 3 exceptionally dry years (Figure 3) and worse in the rest, which are less dry and hence closer to the average climate conditions. After bias correction we worsened the forecast skills these 3 exceptionally dry years, but we improved the skills in the rest. Rather than having higher skills to begin with, in this case the bias correction would have performed better if these 3 dry years would have not been considered, i.e. under less exceptional climate conditions the bias correction would have been more effective.

We think that to find the best bias correction method as well as the best skill score is out of the scope of this study but would be very interesting to look at this in future publications together with the sensitivity analysis mentioned above. We will include this in the section 4.1 Limitations and perspective for future research and implementation.

- The Discussion section contains notes and even recommendations on the use of bias correction. In my view, the poor performance of the bias correction in this case study, and the fact that, as the authors point out, indeed there is only one particular case study analysed here, do not warrant such discussion on the merits of bias correction. Could the authors reflect on this, and depending on whether they agree or disagree, adjust the Discussion accordingly.

In the manuscript we neither recommend nor reject the use of bias correction. We conclude that more studies are needed to investigate the benefits of bias correction when seasonal hydrological forecasts are specifically used to inform water resource management (lines 392-393). Firstly, because this is a case study and secondly because what a skill score reflects may not be representative of the benefits or costs in terms of forecast value of applying bias correction. We will revise the manuscript to make sure that our conclusions do not sound like a recommendations.

- The Discussion section also recommends use of ensemble (probabilistic) forecasts in operational management, which is supported by the research findings, but then connects this to UK policy recommendations on long-term water resources planning. This I think is a bridge too far, and not needed. I would favour the Discussion to be less broad, and stay focused on the research findings presented (see my detailed comments for specific suggestions on where and how to make the Discussion section more specific). This leads to my next comment.

As a case study we do not aim to make general recommendations but rather bring into attention for future studies and practical applications the importance of some aspects or factors such as the uncertainty consideration. We believe that this reference to planning helps the reader understand that while rarely considered currently in short term management, risk-based approaches attempting to deal with the range of potential future conditions expected are already starting to become standard methods in the industry for long term planning. These are two fields that are strongly linked, where seasonal and long term planning are often the responsibility of the same practitioners/teams within companies, or at least teams that strongly interact, and that (could) apply fairly similar methodologies at different time scales.

- I miss a more in-depth discussion on the forecast skill of the DSP used, and the influence of forecast lead time throughout the analysis chain. The CRPSS results nicely show that only for the first two months the uncorrected forecasts have skill (the bias-forecasts do not have skill). Is this positive skill utilised by the operational water management strategies simulated. Could the authors suggest ways on how to capitalise more on this positive skill, e.g. by using DSP for the first 2-month lead time, and using ESP for months 3 to 6?

The Reviewer suggestion is interesting and potentially worth exploring. But we are not sure we will include it because of need to keep the paper concise and because it is not said that what

brings more skill also brings more value. We believe that a more in-depth analysis of the forecast skills-lead time relationship would need a sensitivity analysis what would fit better in a potential future publication already mentioned above. It's difficult to say a priori how a mixed DSP-ESP will perform. Our results overall suggest that inferring the forecast value from its skill may be misleading, given the weak correlation between the two (at least as long as we use skill scores that are not specifically tailored to water resources management). Running simulation experiments of the system operation, as done in this study, can shed more light on the value of different forecast products (lines 402-406).

- Lastly, to come back to the motivation of the authors to bridge science to practice, I would like to see observed and simulated releases for sample priority scenarios and years. These actual releases throughout a season is what operators will recognise and this will enable a discussion on how and to what extend the use of ensemble seasonal forecasts would lead to changes in operation.

We thank the reviewer for this suggestion. While observed releases may not be representative of the system in study, which is a simplification of the real system, we will consider this interesting suggestion, compatibly with the need to keep the manuscript not too long and with confidentiality issues (the reservoir system data used are property of the water company).

---

## Author Response (AR1)

**Author's Response**

**Throughout this response, the reviewer's text is presented in black, our response in blue**

**Reviewer 1**

This an interesting and timely study into the value of forecasts for improving the performance of a simple water supply reservoir system with an operational trade off between augmentation of stored water through pumping and associated energy cost. The selected case study is appropriately simple and also informative for this type of analysis. Results are quite difficult to follow and key details are omitted from the method. The set of forecasts selected for use in the simulation are also poorly justified. Finally, I feel that the paper attempts to answer too many questions and would benefit significantly from more focus. For example, the analysis of the dynamical forecast product and its failure to provide skill over ESP is an interesting study in its own right, demanding much more in-depth analysis and interpretation than is offered in the paper. The operational section then addresses ESP vs dynamical and the additional question relating to importance of incorporating ensemble uncertainty. The paper would be much stronger if you were to focus on just one of these areas and deliver a more compelling conclusion backed up with in-depth analysis of a specific question. I recommend that the paper would be publishable if significant changes are made to simplify the overall story and provide further method detail as outlined in the comments below.

We thank the reviewer for their overall positive evaluation of our manuscript and the suggestions for improvement. We think the analysis of different scenarios is interesting (and other reviewers also seem to agree) and so we intend to keep it. Nevertheless, we appreciate that the manuscript writing is sometimes overly complex and that some analyses (for instance Fig. 6 in the original manuscript) raise more questions than they answer, so in the revised manuscript we have simplified some aspects of the Results section, in particular we have deleted Fig.6 and integrated some of its content into Fig.7 now Fig.6 in the revised manuscript). We have also modified Figure 4 to accommodate the Reviewers' comments (more below) and edited the manuscript throughout for more clarity.

- It's not clear what optimization framework is used to deal with the forecast ensemble. The deterministic approach using rolling horizon (e.g., ANGHILERI et al., 2016) is quite common and there are very few successful examples in the literature where the full ensemble is used to inform the decision. Please outline exactly how the ensemble is used in your optimization and then justify the approach. If this is a new approach it perhaps needs to be described in its own, separate publication.

In our optimisation framework we minimise the expected values of the two objective functions based on the 25 ensemble members (whereas Anghileri et al. 2016 optimised the objective functions evaluated at the ensemble mean). In the revised manuscript, we have given a clearer explanation of this in Sec. 2.3.3 and in the Formulation of the optimization problem (Supplementary material). Regarding the justification of this approach, as we mention on Lines 431-436 of the revised manuscript, the use of the full ensemble has been proved to improve the forecast value in several studies with shorter lead time (i.e. days instead of months), while deterministic approaches such as the one applied by Anghileri et al (2016) did not show significant value in seasonal forecasts. So in our work we used the full ensemble at seasonal scale and also analysed the impact of the ensemble size, which confirm the value of using the full ensemble (see Section 3.2.2 and Figure 5): as discussed in lines 430-431, RTOS outperforms the current operation when using the ensemble forecasts, but it does not if uncertainty is

removed and the ensemble mean is used. We have clarified this also in the Supplementary Materials, where the equations of the model and of the optimisation problem are shown.

- Given the skill scores achieved for the dynamical forecasts, it's not clear why these were pursued in the operational part of the study. What is the justification for using a forecast product that is demonstrated to be unskillful relative to ESP?

One of the objectives of our work was to explore the skill-value relationship and whether one can extract value from forecasts in support of water resource management even if their skill is still relatively low. This is what led us to pursue the evaluation of DSP forecasts as well as ESP. As we mention in the Discussion (lines 407-408) our results suggest that inferring the forecast value from its skill may be misleading. This study indeed contribute to show that seasonal forecasts can deliver benefits to inform operational decisions even if their skill is low (as often the case in extra-tropical areas, such as the UK), and that under certain scenarios DSP can provide higher value than ESP despite its relatively similar skill.

- I found the results quite difficult to follow, partly because it's hard to keep track of the various operational settings. Why not simplify by showing the Pareto front for each forecast set (as opposed to five schemes with different symbols/colors). This would be both more comprehensive and easier to understand. Also, the emojis in the key figures are not appropriate.

We agree with the reviewer that our Figures are quite dense; on the other hand, showing each Pareto front in a different plot may make comparison across sets more difficult. However, in the revised manuscript, we have modified Figure 4 by adding coloured circles to group points under the same operational priority scenario and dashed lines to link points using the same forecast product.

**Reviewer 2**

The main merit of this paper is the proposal of the methodology. The paper forms a valuable contribution to the methodology of quantifying the value of forecasts, here in terms of water availability and energy cost. Probably the methodology is more widely applicable. Generally, the paper is well written, although sentences tend to be too long and their structure could sometimes be made clearer by repeating some short words. Unfortunately, the conclusions from this paper are not really valuable. The problem with the first two conclusions, namely 1) seasonal forecasts can increase value and 2) ESP is hard to beat, is that they are case specific, as acknowledged by the authors (line 470). The third conclusion (the relationship between forecast skill and value is complex) is a trivial one. Below, there is a quite long list of main points, which the authors have to address in my opinion: information about the observations should be given (p1), any procedure based a scenario or forecasts with more inflow than in the worst-case scenario seems beneficial, e.g. taking the median of the historical years (p2), the methodology should be better explained (ps 3, 5 and 6), Mliters are not a valid unit (p4), there is an issue with the bias correction (p7), different processing for the benchmark and other forecasts is questionable (p8), it is strange that the value of the driest years with DSP and ESP processing is not almost equal to the benchmark (p10) and the first part of the discussion section should perhaps be removed (p9). In my opinion should be published after making the suggested major revisions.

We thank the reviewer for their overall positive evaluation and suggestions for improvement. In the revised manuscript we have tried to shorten long sentences and simplify their structure. We have also addressed the specific points raised by the Reviewer, as detailed below.
As for the generalisability of our work, we agree that the methodology here employed is widely applicable, and we are sharing an anonymised version of the code we developed for other users. We have added the link to this toolkit in Section 4.1. As for the generalisability of the results and conclusions, we do not fully agree with the Reviewer. We believe that case studies are necessary to advance our understanding and they allow in-depth, multi-faceted explorations of complex issues. We think that while the results are case specific the conclusions have more general practical implications. First, the study demonstrates that higher forecast skills do not necessarily translate into higher forecast value in reservoir operation and that seasonal forecasts can deliver benefits to inform operational decisions even if their skill is low. Second, we show that the hydrological conditions and the decision maker priorities can have as much or even higher influence on the forecast value than the forecast skill. Third, the study demonstrates the importance of accounting for the forecast uncertainty and highlight the potential benefits with respect to deterministic approaches.

A section about observations (discharge and meteorological forcing) should be added.

In the supplementary material we have added additional information about observational hydroclimatic data

One of the results of this paper is that by basing the operational procedure on the forecasts, less energy for pumping is used while ensuring similar water availability (statistically over the years), compared to basing operational procedures on the worst-case scenario (driest historical year). It is my impression that any operational procedure based on forecasts or scenarios with more inflow into the reservoirs than in the worst-case scenario leads to less pumping and similar water storage, provided the increases are realistic. The authors confirm this in lines 395-397 for the case of applying a bias correction, which increases the inflows to the reservoirs and hence increases the value of the forecasts. So, the worst-case scenario is possibly easy-to-beat by any scenario with more inflow. Somewhere in the paper (in the discussion section?) the following

points need to be discussed. Can this effect on value of increasing the inflow be generalized? What is the value of the forecasts if the operators base their procedure on the scenario of the year with the median value of the historical inflows? I suggest making a calculation with such a scenario. By the way: are the calculations in the worst-case scenario deterministic?

We choose the worst-case scenario as a forecast value benchmark (instead of the median) as this is representative of the current operation of the system, and thus it enables us to show the potential benefits of using seasonal forecast with respect to the current approach. This scenario is actually not so 'easy-to-beat': our results (Figure 5) already demonstrate that deterministic optimisation under a scenario with higher inflows ("DSP-corr deterministic" in Figure 5) does not beat the worst-case scenario (which is also deterministic). In fact, while "DSP-corr deterministic" improves energy savings, it decreases the resource availability for any decision maker priority. We have clarified these points in the discussion (Lines 429-430).

I did not understand Section 2.1 – 1b and c. These paragraphs need to be rewritten. At this stage this paragraph is too abstract. Perhaps providing an example of each concept (operation objective function, optimizer, set of operational decisions) would help. Perhaps merging Sections 2.1 and 2.3 helps. Moreover, after 1b the "set of operational decisions is determined", so why is the operator again "selecting a set of optimal decisions" in 1c? Perhaps lines 124-126 helped me to understand a little bit of what you try to explain, namely that you use hindcasts to evaluate the performance of RTOS. If this is correct, just write that you use hindcasts to evaluate RTOS and discuss a possible operational application in the discussion section.

In this section we try to represent the process that a reservoir operator would follow. In 1.b the operator obtains a set of possible optimal decisions as a result of the optimization of the reservoir system in response to the forecasted inflows. Given that the optimisation problem has multiple objectives, it does not provide one optimal solution but set of Pareto-optimal solutions, each realising a different trade-off between the conflicting objectives. This is why in 1.c. the operator needs to select, according to their priorities, one of the optimal decisions among the ones obtained in 1.b. We have revised our description in section 2.1 to make the point clearer and we have also added examples (Lines 120-121, 146-148).

Replace all appearances of Ml and Ml per some time unit by m$_3$/s (per day is also ok), the common unit in the hydrological literature, e.g. in Figure 2, 3, 6 and 7.

We have replaced Ml by m$^3$ as suggested.

I did not completely understand 2.3.1. Was river R fed by the outflow of reservoir S1 before the dam of S1 was built? Also, it sounds ridiculous to pump water, that was released by gravity from S1 to R, back to S1. So, is the water in R at the location where it is pumped out of the river partly fed by rivers that are not connected to S1? Is S1 located at lower elevation than D, so the water flow needs to be pumped?

As mentioned in the manuscript, the gravity releases from S1 are used to support downstream abstraction during low river (R) flows/season (essentially in Summer). In contrast, during the high flow season (Nov to Mar), pumped inflows from R to S1 may be operated to supplement the natural inflows to S1. The water pumped out in R is fed by rivers that are not connected to S1. We have further clarified this point in Section 2.3.1 and we have improved the system schematic (Figure 2) to make clear that R is fed by both a natural catchment and the gravity release from S1.

I did not really understand those "rule curves" (lines 173-179). Add a figure with a rule curve. It is not clear to me how the refilling ($U_{R,S1}$) is done. Is the "missing water" immediately refilled or is the refilling spread over time until April 1, using the optimizer? In the latter case, how does the optimizer work? Can you give an example of an operational decision? What level is targeted on April 1? How does the operational procedure work for probabilistic forecasts? Since there is variation in resource availability by April 1 (e.g. in Figure 4), storage is not equal to the target on April 1. Can storage be larger than the target or is all water above the target spilled? Can storage be less than the target? Perhaps only in S2 and not in S1 because water can be pumped into the latter basin?

The rule curve applied in the current operation procedures defines the storage level at which pumps are triggered. By the 1 April the objective is to be at full storage. Water is only spilled when the storage is higher than the reservoir capacity. The rule curve is only applied in the current operation approach (benchmark) and not in the RTOS approach. We have further clarified this in section 2.3.6 (Lines 270-272) and in the Supplementary material.

The method of bias correction is not correct (199-203). The number of years used to compute the multiplication factors differ per target year. I suggest using the common leave-one-year-out-method, i.e. the factor for each target year is computed from the data of all other years, including years later than the target year. Your method suggests that it is not allowed to use data from future years but there is no problem in doing so if different years are independent of each other.

Using the leave-one-year-out method works statistically but it does not represent what the operator could have achieved historically if using seasonal forecasts, because at each simulated decision time-step the operator would have only been able to use data up to that moment, and not future years. Given that our methodology aims to simulate the behaviour of the operator and the operational decision-maker process, we must assume that the operator can only have access to past data and hindcasts for the bias correction. We have clarified this point in Section 2.3.2 (Lines 207-208).

Line 264 "but with three main variations": Why do you treat the benchmark differently? This implies that if the forecast is equal to the benchmark, the forecast value differs, which seems undesired.

We treat the benchmark differently because it represents the current operation (Lines 263-266) procedures and we aim to assess the potential of using a real-time optimization system informed by seasonal forecasts in place of current procedures (Lines 102-103). It is virtually impossible that the forecast is equal to the benchmark because it is not possible that the ensemble members are all equal to the worst-case inflow sequence.

The first general lesson in the discussion is "First, we found that the use of bias correction to improve the skill and value of DSP forecast is less straightforward than possibly expected" (lines 381-382). I do not agree. Such an expectation, namely that the forecast skill generally improves due to bias correction (is that the expectation?), just does not exist. Your study indeed confirms that this is a naive expectation. So, remove lines 378-393 or reformulate them. By the way: your bias corrections are based on observations of precipitation and temperature and not on the output (hydrological variables!) of ESP forecasts. So, I did not understand the sentences related to ESP in lines 387-388 and 391-392.

Reading the literature, we have the impression that studies tend to show the benefits of bias correction and it is often recommended or even required for impact assessments. Here some examples:

- From Crochemore et al, 2016: "ECMWF forecast **skill is generally improved** when applying bias correction"
- From Ratri et al (2019): "**Uncorrected meteorological forecasts are not suitable** as direct input for quantitative models, such as those used in agriculture and water management (Schepen et al. 2016). The bias **should be corrected** because it can lead to significant errors in impact assessments (Murphy 1999)." https://doi.org/10.1175/JAMC-D-18-0210.1
- From Schepen et al. 2016: "GCM forecasts suffer from systematic biases, and forecast probabilities derived from ensemble members are **often statistically unreliable**. Hence, it is necessary to postprocess GCM forecasts **to improve skill and statistical reliability**." https://doi.org/10.1175/MWR-D-13-00248.1
- From Zalachori et al 2012: "To **improve the quality** of probabilistic forecasts and provide **reliable estimates** of uncertainty, statistical processing of forecasts is **recommended** (Schaake et al., 2010)" https://doi.org/10.5194/asr-8-135-2012
- From Jabbari and Bae 2020: "Numerical weather prediction (NWP) models produce a quantitative precipitation forecast (QPF), which is vital for a wide range of applications, especially for accurate flash flood forecasting. Since NWP models are subject to many uncertainties, the QPFs **need to be post-processed**. The NWP biases should be corrected **prior to their use as a reliable data source** in hydrological models." https://doi.org/10.3390/atmos11030300

We have clarified this expectation that the forecast skill generally improves due to bias correction by citing in the Introduction several studies such as the ones above (Lines 71-73).

We agree that the sentence on lines 387-388 (original manuscript) was badly formulated and we have now replaced it by: "However, the result points at a possible intrinsic contradiction in the very idea of bias correcting based on climatology." What we aimed to communicate in this paragraph is that since both bias correction and ESP forecast are based on climatology, the bias corrected DSP forecast skills tend to become closer to ESP skills. However, ensuring this skill level with bias correction (Crochemore et al. 2016) may not be the best approach especially under conditions significantly drier or wetter than climatology, which are likely the ones when water managers can extract more value from forecasts. We have further clarified this point in the reviewed manuscript (Lines 386-397)

It is strange that the increase in the value of the system with DSP or ESP forecasts relative to the value of the system based on the worst-case scenario is highest in the driest years (e.g. lines 408-409), while those driest years resemble the worst-case scenario more than the other years. You need to explain this.

The benchmark tends to pump more water during the driest years because the lower storage level is more likely to cross the rule curve and trigger the pumped inflows. This explanation has been now included in the discussion (Lines 414-416).

**General text points**
1) The authors often use the term "bias correction" without mentioning what is corrected. As far as I understood, the forcing of the hydrological model is corrected but if you do not repeat mentioning this now and then, it is confusing because the

output of the hydrological model, i.e. inflow to the reservoirs, can also be bias-corrected. So, replace at numerous places "bias correction" by "bias correction of the meteorological forcing" or "bias correction of the forcing".

This has been corrected accordingly

2) In general sentences are too long, making the manuscript difficult to read. So, shorten sentences where there is an opportunity and make the structure of long sentences clearer, especially by adding some words in sentences with "and". I made some suggestions below (e.g. 15-16 in the abstract).
In the revised manuscript we have tried to shorten long sentences and simplify their structure.

3) "Uncertainty (considerations)" is used to discuss effects of ensemble size and the probabilistic nature of the forecasts. Replace throughout the paper the vague term "uncertainty (considerations)" by the more explicit terms "ensemble size" and "probabilistic/deterministic nature of the forecasts".
We have clarified wherever appropriate that uncertainty in forecast is represented through an ensemble.

**Minor points**
16 Insert "to" before "other factors". Changed accordingly

17 "Some of these factors" is too vague. Write which factors have a significant correlation with forecast value (see point Figure 6 below). Corrected accordingly

24 Add reference to endorse the statement that climate variability is increasing. A new reference has been added

44-45 Replace "it provides" by "they provide" and add "that they" before "reflect". Changed accordingly

54 I miss the logic behind "i.e. ESP …". Replace this part of the sentence or clarify the logic. We have removed it

55-56 Reorder sentence to "The possible improvements of supply-hydropower systems operation due to the use of ESP were assessed by Alemu …" Changed accordingly

69-71 Remove this sentence: this distracts too much. We have kept this sentence because it raises an important point that is later discussed, is bias correction necessary?

78 weaker compared to ….? We have added "than in hydropower production or flood management systems"

83-92 Remove these sentences about some of the many existing metrics. It is not efficient to read about metrics not used in this paper and the metrics used in this evaluation are introduced 2.3.5. We have kept these sentences because they raise an important point that is discussed in this study, i.e. inferring the forecast value from its skill may be misleading and the need for skill scores better tailored to the purpose of the studied system, e.g. such as water resources management

94 Replace "this" by "the". Changed accordingly

99 Replace "simulate and compare" by "assess". Simulate performance sounds strange and it is not clear from the rest of the sentence what is compared with what. Changed accordingly

151 Insert "diagram" after "Pareto front". Changed accordingly

163 Consider removing all text about two companies. It is irrelevant for your story while it is making your story more complex. This text has been removed, and water company has been replaced by system operator.

168 Insert "(R)" after "river" Added accordingly

188 Remove period. Removed

198 Did you also use a multiplicative factor for temperature? No, we use an additive factor, we have corrected this in the text

211 $U_{S1,D}$ is also a pumped water flow according to Fig. 2. Yes, that's right, as described in section 2.3.1

214 Replace "The first objective function" by "Pumping savings" and "The second function" by "Resource availability". Changed accordingly

219 Replace "the 15%" by "only 15%". Changed accordingly

227 Rephrase sentence as follows: "They represent five different trade-offs of operational priorities, according to their relative importance" The sentence has been rephrased as follows: "They represent five different trade-offs of operational priorities, according to the relative importance given to each performance objective"

237 Remove sentence. Removed

247 Remove "and for a given lead time". The role of lead time comes some sentences below. Removed

251 Replace "lead time" by "range of the lead time (we use monthly ranges)" and "CRPS values" by "individual CRPS values". Changed accordingly

252 Replace "CRPS" by "individual CRPS values". Changed accordingly

253 I suggest to replace "mean error" by "discharge bias" since bias in the common word for mean error and the addition of "discharge" helps to distinguish this bias from that in the forcing. I also find the equation redundant. Just write that the bias is the difference between the means of the forecasts and the observation over all ……." We have kept the original terminology, i.e. "mean error", because we believe that using the term "discharge bias" may infer that it can be corrected with bias correction.

262 Add "(1975-76)" after "drought on records". Added

Figure 3 Is this the sum of the inflows to both reservoirs? Are these results for the whole year or a specific part of the year? In the legend of the lower panel "2006" should be replaced by

"2016". This has been clarified in the Fig 3 caption. 2006 is correct, as mentioned in the caption the 3 particularly dry winters are represented and one of them corresponds to 2005-2006.

334 and 338 Replace "Figure 6" by "Figure 5". Changed accordingly

Figure 6 If I just look at these graphs, I get the impression that there is no significant relationship in any of these graphs. However, according to your p-values relationships are significant at the 90% confidence level in panels b and e. Is the calculation of the p-values correct? Or are those low p-values due to using the Spearman coefficient instead of the Pearson coefficient? I think you should use Pearson unless you have good reasons to use Spearman. We have removed this figure and any reference to correlation or p-values.

344 Replace "skill" now and then by "forecast skill", to remember the reader what type of skill this is. This part has changed in the new version of the manuscript

358 Replace "year" by "years". Changed accordingly

395 Replace "reduce" by "reduces". Changed accordingly

399 "improvement of forecast accuracy in some direction". What do you mean by "accuracy"? For me this is something like the root-mean-square-error, which means that there is only one desired direction, namely towards 0. Do you mean something like "a change towards either higher or lower values can be more valuable than a change in the other direction"? We agree the sentence was confusing and we have now removed it and revised the paragraph.

411 Is "Initial storage (total storage value)"equal to "Initial storage" in Figure 6? For clarity, be consequent in the use of specific terms. Moreover, panels c and h in Figure 6 do not show a significant correlation (p-values of 0.21 and 0.80). As mentioned above this figure has been removed

428 Remove ")" Removed

447 Replace, for clarity, "seasonal forecasts" by "seasonal meteorological forecasts". Changed accordingly

465 Insert "of" before "the institutional" Changed accordingly

466 Insert "of" before "the most" Changed accordingly

478-479 Replace "but also the methodology in the first place" by "but in the first place by the methodology". Changed accordingly

This paper addresses comprehensively a topic that increasingly requires investigations: the value of seasonal forecasts for real-life applications. Until recently, many studies have worked on assessing or improving forecast skill, but still few manage to link this skill to value. In addition to being innovative, this study investigates the issue through different uncertainty lenses which makes it a very strong contribution to the field and results in valuable findings both for the scientific community but also for water management stakeholders. Overall, the paper, its ideas, structure and methodology are of high quality. However, additional information, for example about the data used and the forecast skill evaluation, would be necessary to support the analysis. In addition, I have some concerns about the validity of the results on linking skill and value due to the chosen methodology. These are detailed hereafter.

We thank the reviewer for their kind words.

The paper would benefit from a Data section, presenting for instance the data used as reference in the bias correction, the inflows used as reference in the forecast evaluation, or the demand model/observations.

We have added a section titled 'Observational hydrological data' in the Supplementary material (as other Reviewers commented that the manuscript is too long already).

Some additional information would be needed in Section 2.2 on the forecast skill evaluation. More specifically, (1) In Figure 2, two inflows feed the two reservoirs (Is1, Is2), which inflow is being considered when evaluating forecasts? (2) Which time period is used to evaluate the forecasts? In Figure 8, November to April is shown, but in the forecast methodology (Sections 2.2 and 2.3.2) or in Figure 3, no specific time period is mentioned. I would recommend mentioning these two points in 2.1/2a.

The inflow to S1 is used to evaluate the forecast skills from Nov to Apr. We have clarified and mentioned these two points in the case study section (2.3) and in particular in Sec. 2.3.5 (we would rather keep sections 2.1 and 2.2 as general as possible because this methodology is meant to be generalizable and applied by others).

Since the goal of the paper is to assess the added value of dynamical seasonal forecasts for water management, and since authors evaluate the skill and performance of these forecasts against observations, it would be important: (1) to add information about how HBV was calibrated and setup for the area, or at least, to mention its performance (mean error) in simulating past inflows to the reservoirs (giving the possibility to make a parallel with the results in Figure 3); (2) to mention even briefly the hydrological regime upstream the reservoir system, as well as the interannual variability, which will define the added value of a method like DSP over ESP.

We have added this information in the 'Observational hydrological data' section of the Supplementary material.

I have concerns about the methodology chosen to link value and skill, and therefore about some of the subsequent results (first two paragraphs of Section 3.2.3). The authors are trying to link a skill obtained from comparing dynamic forecasts with forecasts based on past climatology (1981-20XX), with a value obtained from comparing dynamic forecasts with a worst-case scenario (1975/1976). On the one hand, the skill is based on the comparison of DSP-corr with ESP, meaning that its value solely indicates the gain in performance from using ECMWF SEAS5 instead of historical precipitation and temperature. On the other hand, the value is obtained by

comparing DSP-corr with a benchmark based on a worst case scenario (1975-1976). These choices result in skills and values that cannot be compared. Instead, the skill computed in the paper could be related to the gains/losses in value between DSP-corr and ESP (difference between the blue and green bars of Figure 7). Reversely, the value computed in the paper could be compared to a skill whose benchmark would be the performance obtained by using the worst case scenario (1975-76) as forecast in all years. This is a major point that should be addressed prior to publication to ensure the validity of the conclusions.

We understand the reviewer concerns and we agree that there is somehow an inconsistency in the definition of the skill and the value, given their different benchmarks. However, we think it is not easy to resolve such inconsistency. In fact, we use ESP as a benchmark for the forecast skill because this is the standard practice in the literature (Pappenberger et al., 2015, Harrigan et al., 2018) as it is more likely to demonstrate the "real skill" of the hydrological forecasting system (lines 253-255), whereas the worst case scenario is never applied as a forecast skill benchmark and it is not likely to demonstrate the "real skill". As for the forecast value, we use the worst-case-scenario as benchmark because it is the scenario applied in the current operation approach and hence it is a benchmark that can demonstrate the "real value" of moving away from that approach towards using seasonal forecasts (lines 265-266). We would thus be reluctant to change any of these benchmarks, as they are appropriate for their different purposes.
However, we agree with the Reviewer that this inconsistency should prevent one from directly comparing the numerical values of the forecast skill and value. So, we have now removed Figure 6 of the original manuscript, which incorrectly suggested that such value-by-value comparison can be drawn. On the other hand, we think one can still compare the ranking of forecast products induced by the forecast skill with the ranking based on the value, which is the main point of our work. This is what we refer to when discussing the "forecast skill-value relationship" (line 98). With this clarification, we think the year-by-year analysis of the different optimisation results (Figure 7 in the original manuscript, now Figure 6 in the revised version) still provides interesting insights, and supports the main conclusion that "the relationship between the forecast skill and its value for decision-making is strongly affected by the decision maker priorities and the hydrological conditions in each specific year" (lines 491-492).

**Specific comments**
L45: "and reflects the risk-averse attitude"

L68: "to continue increasing in coming years"

L75-76: More literature review would be needed on past works on seasonal (climate or hydrological) forecasts value. Here are a few examples to consider:

Bruno Soares, Marta, Meaghan Daly, and Suraje Dessai. 'Assessing the Value of Seasonal Climate Forecasts for Decision-Making'. *WIREs Climate Change* 9, no. 4 (2018): e523. doi:10.1002/wcc.523.

Giuliani, M., L. Crochemore, I. Pechlivanidis, and A. Castelletti. 'From Skill to Value: Isolating the Influence of End-User Behaviour on Seasonal Forecast Assessment'. Hydrology and Earth System Sciences Discussions 2020 (2020): 1–20. doi:10.5194/hess-2019-659.

Parton, Kevin A., Jason Crean, and Peter Hayman. 'The Value of Seasonal Climate Forecasts for Australian Agriculture'. *Agricultural Systems* 174 (2019): 1–10. doi:10.1016/j.agsy.2019.04.005.

Thank you for the interesting references. In this part of the literature review we refer to the lack of pilot studies demonstrating the value of seasonal forecast products in UK and Europe but it was not very clear in the manuscript so we have further clarified this in the new version of the

manuscript. We make reference to past general studies on seasonal forecast value in reservoir operation in the 2nd paragraph of the Introduction.

We thank the reviewer for the references. The Bruno Soares et al (2018) study has been included it in the Introduction as well as Giuliani et al. (2020). The latter is the first pilot application in Europe demonstrating the value of seasonal forecast products, in this case ECMWF, that we have found in the literature. It must be noted that this study is still under review.

The study by Parton et al (2019) is a review of the use of seasonal forecast in agriculture in Australia. However, in the context of our study, we could not find any reference using seasonal forecast for reservoir operation.

L107: In this sentence, the authors seem to assume that their results could be generalized to extra-tropical regions and that the potential for use is region-dependent. However, as the authors state, results (forecast value) are in fact highly dependent on the investigated system, and therefore this sentence may sound a bit too ambitious compared to the paper objectives.

We have deleted the reference to extra-tropical areas.

Section 2.1: I found this section very clear and helpful.

We thank the reviewer for this positive comment.

L118: Here, I was wondering what type of demand model was used. It is later in the discussion that the authors mentioned that the demand is based on observations. This would be worth mentioning earlier in the text, and would fit in a Data section.

This is mentioned in section 2.3.3. As we would like to keep this section as generalizable as possible we mention this later, in the Case study, in particular in section 2.3.3.

L142 (2.a): More information would be needed at this stage about the forecast skill evaluation, for example by mentioning the reference and the benchmark used.

Since we would like this section to be used by others and hence to be as generalizable as possible, we have deleted this reference to the specifics of our case study (explained in section 2.3) in 2.a as well as 2.b

L160-161: The notation for units is not consistent between the text (Ml) and Figure 2 (ML).

This has been corrected. In the new version of the manuscript we use m3 instead of ML

L176: Did you mean a "reasonable chance"?

Yes, thank you

L182: Please consider adding the following reference for ECMWF SEAS5:

Johnson, S. J., Stockdale, T. N., Ferranti, L., Balmaseda, M. A., Molteni, F., Magnusson, L., Tietsche, S., Decremer, D., Weisheimer, A., Balsamo, G., Keeley, S. P. E., Mogensen, K., Zuo, H. and Monge-Sanz, B. M.: SEAS5: the new ECMWF seasonal forecast system, Geoscientific Model Development, 12(3), 1087–1117, doi:10.5194/gmd-12-1087-2019, 2019.

Thanks, this reference has been added.

L182-183: I suggest writing "The ECMWF SEAS5 hindcast dataset" because it is the hindcast dataset that includes 25 members, while the real-time operational ECMWF SEAS5 has more members.

Changed accordingly.

L187: There are two points on this line.

Corrected

L195: This statement needs to be refined. Linear scaling and distribution mapping may lead to similar results in terms of bias removal, but they will have very different results on the forecast themselves, e.g. linear scaling will just shift the ensemble, while distribution mapping will have a different correction for each member, resulting in larger impacts on the spread.

We have added this clarification

L196-198: Usually a difference approach is applied to calculate the correction factor for temperatures (see e.g. Lucatero et al. 2018, Teutschbein and Seibert 2012). Similarly the correction factor is added/subtracted to correct the raw forecasts. If a ratio approach was applied for temperatures, I would suggest adding a short justification of this choice. For instance, how are negative temperatures handled?

Lucatero, D., Madsen, H., Refsgaard, J. C., Kidmose, J. and Jensen, K. H.: On the skill of raw and post-processed ensemble seasonal meteorological forecasts in Denmark, Hydrology and Earth System Sciences, 22(12), 6591–6609, doi:10.5194/hess-22-6591-2018, 2018.
Teutschbein, C. and Seibert, J.: Bias correction of regional climate model simulations for hydrological climate-change impact studies: Review and evaluation of different methods, Journal of Hydrology, 456–457, 12–29, doi:10.1016/j.jhydrol.2012.05.052, 2012.

We used an additive factor in the case of temperature but we did not mention this in the manuscript. We have clarified this in the new version of the manuscript

L206: "with what was done"

Corrected

L208: The ensemble size has an impact on the CRPS (Ferro et al. 2008) and therefore, on the skill when the benchmark has a different ensemble size than the evaluated system. This could impact the results presented in this paper, and I would suggest adding comment about the impact of the varying ESP ensemble size on your skill results.
Ferro, C. A. T., Richardson, D. S. and Weigel, A. P.: On the effect of ensemble size on the discrete and continuous ranked probability scores, Met. Apps, 15(1), 19–24, doi:10.1002/met.45, 2008.

We thank the reviewer for the comment and reference suggested. However, the ensemble size is very unlikely to have a considerable effect on the results of this study because a) the ESP forecast quality in this study is too limited and the time range too short to have an important effect on the forecast skills and b) the results already show that the forecast value is fairly insensitive to changes in the ensemble size (Figure 5) unless a very low number of members (5 or less) are considered. Nevertheless, it would be an interesting question to explore by means of a sensitivity analysis in a future publication.

Section 2.3.5: I would suggest presenting the CRPS (L245-252) before presenting the CRPSS (L239-244) since it would define the CRPS notation before using it in the skill equation. There are also inconsistencies between the first two equations (CRPS and CRPSS) and the third one (mean error), such as the notation for the observations (*y* vs *I*$_{Obs}$), the notation for the system (*Sys* vs *Syst*), the mention of the lead time only in the third equation but not in the previous ones.

We have corrected the inconsistencies and we have also changed the order and now CRPS is presented before CRPSS as suggested by the reviewer

L237-238: This sentence is repeated twice.

We have removed the duplicated sentence

L243: By choosing ESP as a benchmark, you also make the decision of only analysing the added skill from using dynamic weather forecasts over meteorological history, it would be worth mentioning.

*We have mentioned this as suggested by the reviewer.*

L256: Do I understand correctly that the mean error is computed over a time window that varies with the lead time you consider, for example to evaluate lead time 6 weeks, you average the mean error from week 0, 1, … to 6? Is the same calculation method applied when computing the CRPS?

The average CRPS or mean error for a given lead time is equal to the average of the individual CRPS or mean error values obtained for the forecasts published across the time frame. We have clarified this in the revised manuscript.

L279: I suggest replacing "gets lower" with "gets larger in absolute value".

Here we refer to DSP-corr which gets lower and not larger for longer lead times

L334: The reference should be to Figure 5 instead of Figure 6.

Corrected

L358: "two specific years"

Corrected

L381-393: In this paragraph, conclusions are likely only valid for linear scaling, and not for any bias correction. I would suggest being more specific throughout the paragraph.

We have mentioned that in particular we refer to linear scaling

L446: "that should be kept in mind"

Corrected

Figure 6: I wonder why, for each of the five operation scenarios, the authors do not display both DSP-corr and ESP in the same graphs.

This figure has been removed in the revised version of the manuscript after comments from other reviewers

Figure 6b and 6g: Shouldn't the absolute mean error be used to assess the correlation between the increase of resource availability and the performance, since both high positive and low negatives reflect poor performance?

This figure has been removed in the revised version of the manuscript after comments from other reviewers but we wanted to keep the sign of the error because we have observed that underestimation has different impacts on the forecast value as compared to overestimation.

Figure 7: It could be interesting to see the mean error and the CRPSS in this figure, even if they are not correlated.

We have added both.

Supplementary material – Figure 8: In the legend, I could not remember that 1975-1976 was the worst-case scenario. It could be worth mentioning it again at the end of the caption.

We have specified this in the end of the caption too

Supplementary material – Figures 9 to 13: The CRPSS is used to try and explain the gains in value from using ESP. Here, it is not clear which benchmark the performance of ESP was compared to in order to compute the skill, if it is indeed the CRPSS of ESP being shown. If not, then please refer to the fourth general comment.

These figures have been removed in the revised version of the manuscript after comments from other reviewers

Dear authors,

Thank you for this interesting research, written up in a well-organised and clear paper. Overall, I have not hesitation to recommend your paper for publication. I do agree with you, that this kind of research, with continuous simulation of operational water management to test new information sources, methods, or strategies, is valuable for science and in particular for bringing findings forward to practice in an informed and iterative way.

I appreciate in particular your Conclusions section, and clear description of data used, methodology, and presentation of results.

We thank the reviewer for their kind words.

General comments
I have the following main comments:

- The authors assumed the water demand to be known in advance and to be equal to the observed reservoir releases (line 220). This may be an important assumption. If more than needed was pumped-in for storage, this would perhaps also lead to releasing more than the actual demand. Could the authors reflect on this? The actual releases are the result of the current water management priorities, which focus on water resources availability. Could this have led to the forecast value also being maximum for the rap scenario's? Could the authors address this with a limited sensitivity analysis, varying water demand? (if time, sensitivity analysis of other aspects would be interesting as well, e.g. towards the set-up and settings of the NSGA optimisation experiment.)

The release data that we use are only the controlled releases (from the outlet tower) and do not include spills, so we believe that our assumption that they reflect the demand is reasonable, and they should not exceed the actual demands, as suggested by the Reviewer. Moreover, in our experiment we have only simulated the refill period during winter when demands are fairly stable and predictable, hence also the assumption that demand are known in advance should not be too limiting. Finally, the forecast value is quantified with respect to the benchmark and both benchmark and DSP under all the scenarios assume the same demand, so it is unlikely that changing the demand is going to have an influence on how better DSP does with respect to the benchmark. We have clarified these points in section 2.3.3 and in the Details of the optimization (Supplementary material). This said, we agree that one could test the influence of the demand assumption through a sensitivity analysis, but we would leave this experiment to future studies, especially as the manuscript is already quite long (and other Reviewers asked to shorten it already). We have thus only mentioned the point in the Discussion of future research (Lines 462-467).

- To my view, the results show that the bias correction applied, did not work in this particular case study, Figure 3 (only changed sign of MAE from under- to over-prediction, as authors also indicate in line 364). Could the authors reflect on this in their section on limitations of the research? What may be the reason? Could other bias correction methods work better or is this not to be expected (e.g. perhaps higher forecast skill is needed to begin with, for post-processing methods to be effective)? The poor performance of the bias correction also connects to the following comment.

The main reason for the bias correction to fail is the DSP forecast was already doing relatively good in terms of skills in 3 exceptionally dry years (Figure 3) and worse in the others, which are closer to the average climate conditions. After bias correction, the forecast skills are worsened

in these 3 exceptionally dry years, but improved in the others. We have included this explanation in the Discussion. We think that to find the best bias correction method as well as the best skill score is out of the scope of this study but would be interesting to look at this in future works together with the sensitivity analysis mentioned above. We have included this in the section 4.1 Limitations and perspective for future research and implementation (Lines 462-467).

- The Discussion section contains notes and even recommendations on the use of bias correction. In my view, the poor performance of the bias correction in this case study, and the fact that, as the authors point out, indeed there is only one particular case study analysed here, do not warrant such discussion on the merits of bias correction. Could the authors reflect on this, and depending on whether they agree or disagree, adjust the Discussion accordingly.

We do not intend to recommend nor reject the use of bias correction. We conclude that more studies are needed to investigate the benefits of bias correction when seasonal hydrological forecasts are specifically used to inform water resource management. We have now revised the manuscript to make sure that our conclusions do not sound like recommendations (Lines 395-397)

- The Discussion section also recommends use of ensemble (probabilistic) forecasts in operational management, which is supported by the research findings, but then connects this to UK policy recommendations on long-term water resources planning. This I think is a bridge too far, and not needed. I would favour the Discussion to be less broad, and stay focused on the research findings presented (see my detailed comments for specific suggestions on where and how to make the Discussion section more specific). This leads to my next comment.

As a case study we do not aim to make general recommendations but rather bring into attention for future studies and practical applications the importance of some aspects or factors such as the uncertainty consideration. We believe that this reference to planning helps the reader understand that while rarely considered currently in short term management, risk-based approaches attempting to deal with the range of potential future conditions expected are already starting to become standard methods in the industry for long term planning. These are two fields that are strongly linked, where seasonal and long term planning are often the responsibility of the same practitioners/teams within companies, or at least teams that strongly interact, and that (could) apply fairly similar methodologies at different time scales.

- I miss a more in-depth discussion on the forecast skill of the DSP used, and the influence of forecast lead time throughout the analysis chain. The CRPSS results nicely show that only for the first two months the uncorrected forecasts have skill (the bias-forecasts do not have skill). Is this positive skill utilised by the operational water management strategies simulated. Could the authors suggest ways on how to capitalise more on this positive skill, e.g. by using DSP for the first 2-month lead time, and using ESP for months 3 to 6?

The Reviewer suggestion is interesting and potentially worth exploring in future studies. We have not included it because of the need to keep the paper concise and because it is not said that what brings more skill also brings more value. We believe that a more in-depth analysis of the forecast skills-lead time relationship would need a sensitivity analysis what would fit better in a potential future publication. It is difficult to say a priori how a mixed DSP-ESP would perform. Besides, our results overall suggest that inferring the forecast value from its skill may be misleading, given the weak correlation between the two (at least as long as we use skill scores that are not specifically tailored to water resources management). We have included this in Sec. 4.1. as a suggestion for future studies (Lines 462-467)

- Lastly, to come back to the motivation of the authors to bridge science to practice, I would like to see observed and simulated releases for sample priority scenarios and years. These actual releases throughout a season is what operators will recognise and this will enable a discussion on how and to what extend the use of ensemble seasonal forecasts would lead to changes in operation.

We thank the reviewer for the interesting suggestion. We cannot, however, publish the company data on reservoir releases. The value of the data may also not be representative also of the simplified system in study, which is part of a broader conjunctive use system, where releases may be driven by broader resource considerations.

**Detailed comments:**
- line 275: Indeed the bias corrected DSP "improves" skill for longer lead times, but only from negative skill to less negative skill. I would suggest to point that out.

We have pointed this out.

- line 281: Yes, with bias correction there is "some improvements for some lead times", but still with negative skill. Rather than pointing out "some improvements for some lead times", I think it is more relevant to point out here that the bias correction as applied here, in this case study, is not working and even has adverse effect on forecast performance.

We have modified this paragraph to reflect the in summary the bias correction does not produce an improvement in the forecast skills

- line 311: The question is why? Again (See my first general comment, and note that resource availability in the results varies only with 1-2%) this may indicate a constraint set-up, favouring a focus on rao. Please reflect on this.

Rather than a constraint set-up, this can simply be explained by the reduction of the pumping costs after bias correction, which is a consequence of the overestimation of the inflows. We have now included this explanation to rap and *bal* Pareto dominating the benchmark which can be also applied to *rao* (lines 400-401).

- line 382: "Our results show that on average bias correction slightly improves the DSP forecast skill (as measured by CRPSS and mean error)". I do not agree here. When looking at the results, also on average, bias correction reduced the CRPSS for the first 2 months lead time where the DSP had skill (bias correction made skill less negative for further lead times, but still negative), and it changed the sign of the average error but did not reduce it, so bias correction was not working well.

We agree on this with the reviewer and we have modified this paragraph accordingly.

- line 392: I agree with the authors. Based on this particular study, not much can be discussed on the merits of bias correction. Instead I would recommend to focus discussion in the paper more on the skill of the ensemble DSP (slightly positive for the first 2 months), how and why this is or is not being used in the water system operations simulated (see my last two General comments above).

We thank the reviewer for this interesting suggestion. As mentioned above, we have included the possibility of combining DSP with ESP as a way to improve seasonal forecast as a suggestion for future studies.

- lines 399-404: While I agree with this statement in general, I do not think the research presented provides sufficient supporting evidence. Nor is it a surprise or a new finding.

It is true that this cannot be generalized but we believe that the results of this particular study support the need to further investigate the adequacy of different skill scores to evaluate the forecast value for water management purposes. To our knowledge, the literature on forecasts evaluation often overlook the issue of defining scores that are specifically tailored to a particular purpose.

- lines 413-416: This is quite a strong recommendation not substantiated with any analysis/numbers on what these 'costs' are. The authors could consider leaving this out.

This was not meant to be a recommendation but a hypothetical and explanatory scenario where one product could be preferred over the other. As mentioned above, we do not aim to make general recommendations but rather bring into attention for future studies and practical applications the importance of some aspects or factors. We have revised the manuscript to make sure that they do not sound like recommendations.

- lines 433-436: I am not sure if the link with Long-term water resources planning is appropriate, and also I think it is not needed to make the case of risk-based operation on the basis of ensemble (probabilistic) forecasts. The results of your study do show this nice enough. The authors could consider leaving this out.

As already mentioned above, we believe that this reference to planning may be useful for some readers (particularly UK practitioners) who are maybe familiar with risk-based approaches in long-term planning, and point them to the fact that similar concepts may be usefully applied in short-term management.

- lines 458-467: Yes, developing such toolkit and making available to the water management organisation is very valuable. Indeed question here would be to what extend the toolkit is customised to the specific case study, and how much time/effort customisation to a new case study would require. Could the authors reflect on this in the text?

This toolkit was tailored to this study case. It is an interesting and relevant question that we are still not able to answer but we are now testing the use of the toolkit as a mechanism to support knowledge transfer to practitioners and to evaluate how easily they can customise it, and this will be the focus of a future publication. We have clarified this in section 4.1 (Lines 475-478).

- line 475: DSP is now more readily available from international weather forecast centres and more easily processed, such that this by-sentence on "ESP being more easily derived", in my view is perhaps becoming less relevant. Also because, as the authors describe, they have provided a Toolkit for ease-of-use.

From our own experience and through our collaboration with practitioners at water companies, downloading and post-processing the seasonal forecasts still needs a considerable level of expertise and while weather forecast centres, such as ECMWF, are gradually facilitating the access to the data, the process of bias correction is still quite difficult. Not only because they still do not provide with such tools but because first we need to decide whether applying bias correction or not and also select (and understand) the most adequate bias-correction method. And as we have seen in this study this is still not clear. This has been added to the Discussion (Lines 418-422)

Please see for suggested technical changes (editorials) the annotated pdf.

Apart from the pdf with the reviewer comments we couldn't find an annotated pdf.

Thank you and with best regards.

**List of all relevant changes**

Dear editor,

Thank you for the opportunity to respond to the reviewers' comments and submit a revised manuscript.

The reviews were positive overall and very helpful to improve the manuscript. However, sometimes the reviewers asked for changes or additions to the paper in different directions so that we found it impossible to accommodate all of them simultaneously. Some reviewers appreciated the applied nature of our work and suggested to give more details about the case study, while another reviewer asked for more details about the methodology; all reviewers suggested further analyses of various aspects of our modelling chain, while also asking for a more concise, shorter paper. Some of these suggestions for further analysis are very interesting and are worthwhile exploring in future publications, but in our opinion are beyond the scope of this paper.

So, we have tried to accommodate as many as possible of the reviewers' comments while also maintaining the focus on the key question of our work (what is the value of seasonal forecast for water resource management?) and mentioned in the Discussion session the further analyses that we think are beyond the scope of this paper.

List of relevant changes. We have:

- edited the manuscript throughout for more clarity
- added more references
- improved the following aspects of the Methodology:
    o clarified the formulation of the optimization problem
    o further justified the use of ensemble modelling instead of deterministic, the use of past data for bias correction method and the use of ESP as forecast value benchmark
- improved the following aspects of the Case study description:
    o the system schematic (Fig. 2)
    o the explanation of how the rule curve operates
    o added information about the observed hydrological data in the Supplementary material
- simplified the Results section:
    o removed Fig. 6 (in the original manuscript), which we realised was making the storyline unnecessarily complex and was raising more questions than it was answering. Some of the content of that Fig. 6 has been integrated into what was Fig.7 (and is now Fig.6 in the revised manuscript).
- improved the Discussion section:
    o further clarified the reasons of why bias correction deteriorates the forecast skills but improves the forecast value and further discussed whether ESP or DSP should be applied
    o added further possible future research.
- shared a link to an anonymised version of the code for application of our methodology.

[revised manuscript text omitted]

---

## Author Response (AR2)

**Authors' Response**

**Throughout this response, the reviewer's text is presented in black, our response in blue**

**Report #1**

I want to thank the authors for addressing previous comments and for their comprehensive replies. I found all replies satisfactory and the changes made to the manuscript significantly improve the quality of the paper. Removing Figure 6 and focusing the analysis of the value in Figure 7 answers my prior concern about the mismatch in terms of benchmarks. I list some additional minor comments below, including a methodological point (point 3) which could potentially be major.

REPLY: We thank again the reviewer for their overall positive evaluation of our manuscript and the suggestions for improvement.

L255 "based on dynamic weather forecasts": I suggest replacing this with "and the added performance of dynamic weather forcings rather than historical ones"

REPLY: We have replaced the original sentence by "and the added performance of dynamic weather forecasts".

L259: In the mean error equation, the averages over the number of members and time steps seem erroneous. The sums from 0 to M and from 0 to T should either be from 1 to M and 1 to T or the divisions on T+1 and M+1.

REPLY: Thanks for picking this up. This has been corrected, now the sum starts at 1 and ends in T and M.

L280-281: Here, it may be that the sentence needs reformulation or it may simply be a matter of terminology; otherwise it seems that you are averaging forecasts of different lead times but same forecast horizon. 1 Jan-1 Apr is a 3-month lead forecast of April 1st, 1 Feb-1 Apr a 2-month lead forecast of April 1st and 1 Mar-1 Apr a 1-month lead forecast of April 1st. Therefore, averaging these will not give a CRPSS for a 3-month lead forecast. Instead, it gives an average performance of the forecasts with a similar forecast horizon (but with different lead times).

REPLY: We agree with the reviewer that this needs to be further clarified. The average skill (CRPSS) for a given time step is the average skill of all the forecasts used from that time to the end of the simulation period (April 1st). This aims to represent the average skills of the forecast that a reservoir operator would have available during the operation process. This also makes the comparison between skills and value (which is also averaged over the simulation horizon) more meaningful. We have further clarified this by revising Section 3.1 (lines 277-283) and Figure 3.

L683: Replace "Solid lines represents" with "Solid lines represent"

REPLY: Corrected accordingly

I have some remaining clarifications on the optimization approach using ensemble members. I understand from the revision that the forecast ensemble is used in the following way: evaluate decision set against each ensemble member separately, compute objective function for each, take expected (mean) value of objective function; optimize decision to minimize this expected value. This approach is typically avoided in multi-stage settings because it fails to account for decision recourse. This is why multi-stage stochastic programming using scenarios trees has been developed in prior studies, e.g.,:

https://www.sciencedirect.com/science/article/pii/S1364815212002770
https://www.sciencedirect.com/science/article/pii/S0309170814001262

Here you appear to have taken a shortcut, yet still achieved reasonable results. Please place your approach in this context and offer some discussion on why your approach works despite neglecting decision recourse.

REPLY: We thank again the reviewer for these comments. We agree with the reviewer that using multi-stage stochastic programming we could achieve better optimisation results and hence an improvement in the forecast value. However, as the Reviewer also pointed out, the simpler approach used here already provides reasonable results. Given the low forecast skill in this case study, we believe that the improvement in the forecast value by a more sophisticate optimisation approach is likely to be modest and possibly not sufficient to justify the increase in computational cost and complexity. In the revised manuscript, we have added a clarification and discussion of this point in the Formulation of the optimization problem (Supplementary material) and in the Discussion (lines 465-466).

I also don't quite understand why you claim that the approach is superior to the ensemble mean. Results in Figure 5 seem to suggest that the deterministic approach lies along a Pareto front for the chosen objectives. From the perspective of energy savings, aren't the deterministic results better than ensemble? I feel that the mean approach and the ensemble approach need to be compared more rigorously.

REPLY: Figure 5 shows that the deterministic approach only improves one objective, Pumping energy cost savings, at the expenses of deteriorating the other objective, Increase of resource availability by 1 April, and this happens no matter the decision priority that we select. While this may be acceptable in scenarios that prioritise energy savings (pso and psp), in the scenarios where resource availability is optimised individually (rao) or prioritised (rap), the fact that this objective is worse than in the benchmark means that using deterministic forecast has effectively no value. We clarified this point in Sec. 3.2.2 (lines 333-338) and in Sec. 4 (lines 423-430).